



# Satellite-based Tracking of Reservoir Operations for Flood Management during the 2018 Extreme Weather Event in Kerala, India

Sarath Suresh[1], Faisal Hossain[1], Sanchit Minocha[1], Pritam Das[1], Shahzaib Khan[1], Hyongki Lee[2],
Konstantinos Andreadis[3] and Perry Oddo[4,5]

[1] Department of Civil and Environmental Engineering, University of Washington, Seattle, 98195, USA

[2] Department of Civil and Environmental Engineering, University of Houston, Houston, 77204, USA

[3] Department of Civil and Environmental Engineering, University of Massachusetts, Massachusetts, 01003, USA

[4] Hydrological Sciences Branch, Goddard Space Flight Center, Greenbelt, Maryland, 20771, USA

[5] Science Systems and Applications, Inc., Lanham, MD, USA

*Corresponding Author:* Faisal Hossain (fhossain@uw.edu)

**Abstract.** In parts of the world characterised by high precipitation and steep topography, hydroelectric dams often play the dual role of power generation and flood control. Improper and uncoordinated management of such dams during extreme and unexpected precipitation events can have disastrous consequences. As such, there exists a growing need for a reliable, transparent, and publicly available reservoir assessment and information system that can help water managers better prepare for such natural events. A fully satellite sensor-based framework offers a potentially viable approach towards this goal. The Reservoir Assessment Tool (RAT 3.0), which utilises high frequency remote sensing-based surface area and reservoir storage estimation alongside hydrological modelled inflow is tested and analysed for the 2018 Kerala floods in India. The effectiveness of RAT 3.0 was gauged by considering how well a fully satellite sensor-based framework was able to capture the rapidly evolving dynamics of the flood and reservoir state. Application of RAT 3.0 in monitoring the state of 19 reservoirs in Kerala during the flood event showed very promising results. In general, RAT 3.0 was found to be able to capture the temporal trend of the reservoir storage and pinpoint the sudden shift in filling or release decisions made by the dam operator. This translated to reliable updating of downstream flood risk in near real-time for improving flood preparedness, even though the absolute magnitudes were sometimes found to be in need of bias correction. A customised form of RAT 3.0 tailored for hydropower dams operating during high precipitation events in mountainous regions is proposed as an outcome of this study.

**Keywords: Floods, hydropower, extreme weather, mountainous terrain, reservoirs, remote sensing**

## 1 Introduction

The role of dams and reservoirs as a critical line of defence against the growing intensification and unpredictability of floods cannot be overstated. Variation in the magnitude and frequency of precipitation events, driven by the overall warming of the planet is widely considered the root cause of this uncertainty (Kharin et al., 2013; Berg et al., 2013 ). Today, more than half of





the world's major river systems are regulated by dams (Nilsson et al., 2005) with an estimated total water impoundment capacity of ~10,000 km$^3$ (Chao et al., 2008). The International Commission on Large Dams, a leading database on the world's dams, identifies 2539 large dams as having the sole purpose of flood control and over 4900 dams operating in a multi-purpose role that includes flood control. However, there exists an intrinsically competing combination of hydropower dams that also

operate in the capacity of providing flood control. It is in the interest of these dams to maintain as high a water level as possible to maximise power generation which is in direct conflict with the objective of flood moderation. Such hydropower dams that provide flood control are typically found in regions characterised by high rainfall and steep mountainous topography, with a high dependency on hydro-electric power to meet energy needs.

Roughly 25% of the Earth's land surface is classified as mountainous, referred to as 'natural water towers,' contributing to a

disproportionately higher (~60%) of total discharge in river basins (Viviroli et al., 2007). Hydropower dams constructed across the numerous rivers in such areas generate over 4300 TWh of power constituting nearly 17% of the total electricity produced in the world (IEA report, 2021). The long-standing operating policy in many of these dams is to maintain the water level close to or at the full-reservoir level (FRL) for a major part of the year to maximise power generation (Miao et al., 2016; Ahmad and Hossain, 2020). Thus, the operating rule of these dams, which is a series of predetermined water levels or storage capacities,

act as a guideline for the release and retainment of water and is often fixed with little room for flexibility. Unfortunately, such a rigid and static operating rule-based framework is currently proving ineffective against the mounting unpredictability of precipitation and associated runoffs. In recent years, numerous cases of flooding events that could have been avoided by timely alteration of the reservoir levels have been identified across the world (Kundu & Mothikumar, 1995; Zhang et al., 2014). The closed-access and often classified nature of reservoir operations data, on grounds of national security, is also known to

exacerbate the difficulty in timely disaster management efforts and public evacuation plans (Owen et al., 2020). Furthermore, co-ordinated flood management strategies involving multiple reservoirs operated by different controlling agencies spanning multiple jurisdictional boundaries are nearly impossible without the presence of a near real-time and open data sharing framework. This study emphasizes the pressing requirement for an effective and transparent monitoring system for dams and reservoirs (hereafter used interchangeably) worldwide where hydropower generation and flood control are extensive, such as

in regions characterized by mountainous terrain and high rainfall (Figure 1). The system should be reliable, provide near real-time data on the state of the reservoirs that have to co-optimize flood control and hydropower, and be accessible to the public. Space-borne satellites hold immense potential for monitoring reservoir operations, offering a wide range of valuable information to enhance reservoir operation strategies. Satellites such as the Landsat series from National Aeronautics and Space Administration (NASA), Sentinel constellation from European Space Agency (ESA), the Terra and Aqua from NASA

carrying the MODIS (Moderate Resolution Imaging Spectroradiometer) instrument, GRACE-FO (Gravity Recovery and Climate Experiment - Follow-On) jointly developed by NASA and German Space Agency, and SWOT (Surface Water and Ocean Topography) by NASA and French Space Agency, provides data on the extent of surface water features. Altimetry satellites such as Jason series, Sentinel-3, and the Ice, Cloud and land Elevation Satellite (ICESat), provide accurate water level estimates, which have been used along with remotely sensed water surface areas to derive estimates of reservoir water





storage (Chen et al., 2022; Cooley et al., 2021; Gao, 2015). The water storage variation along with hydrological model-based inflow can be used to solve the water balance equation to infer reservoir releases (Zhong et al., 2020, Bonnema and Hossain, 2017, 2019). Satellite based remote sensing also has the distinct advantage of having global coverage, thus bypassing restrictions due to jurisdictional and administrative data governance policies.

The issue of co-optimizing hydropower and flood control in mountainous regions with high precipitation is a global one, and

hence satellite remote sensing is the only viable and practical way to monitor the reservoir state in an open data sharing framework. Figure 1 highlights various regions around the world that experience high precipitation and has steep topography. Dams located in these regions are often simultaneously utilised for flood control and hydropower generation. Notable locations include the Western Ghats of India, western Norway, the Southern Alps in New Zealand, the Andes Mountains in Peru and Chile, Indonesia, Japan, Guinea and the Pacific Northwest region of the United States and Canada. During extreme

precipitation events, there might occur cases of reservoir management failing to account for the sudden rise in inflow. If the reservoir level is not adequately low enough to anticipate this, then the dams would have to release the water suddenly, often leading to disastrous consequences downstream due to lack of preparation. If the reservoir is used for the purpose of producing hydroelectric power, then the odds of such sudden release are much higher due to the very nature of operation of such dams. Such areas are also prone to intense cloud cover for a significant portion of the year, making optical satellite observations

difficult. Hence, the monitoring system should be capable of tackling the issue of high cloud cover and must incorporate cloud masking techniques.

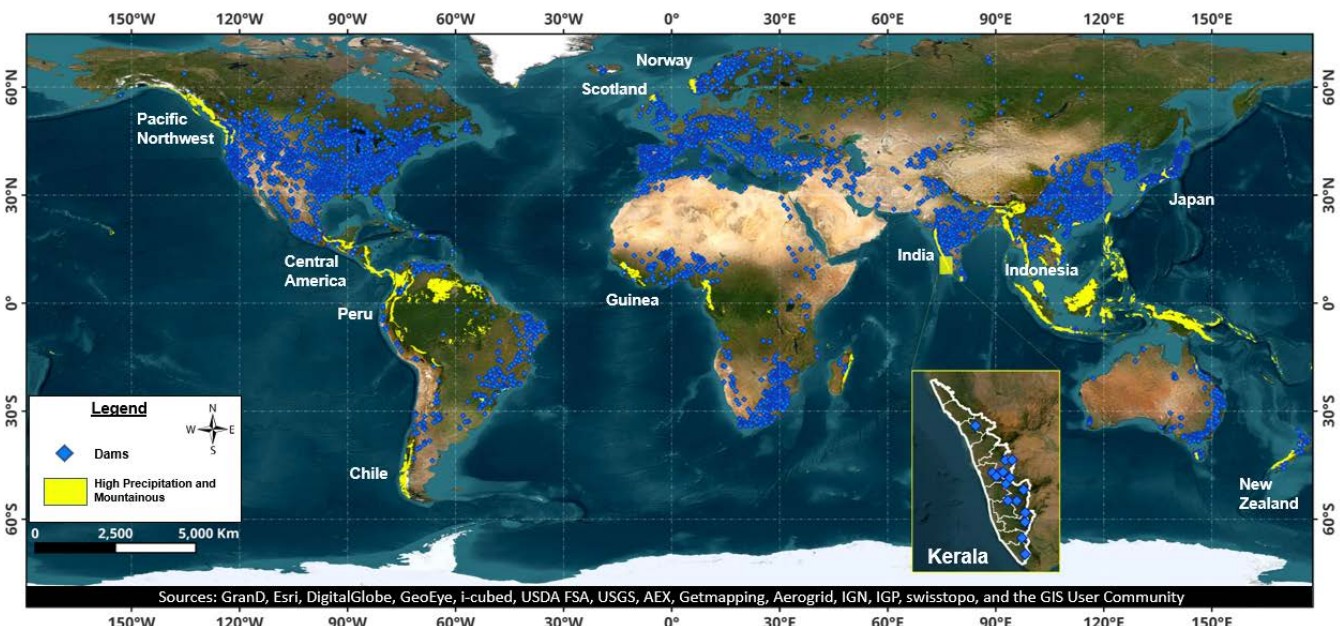

Figure 1: World map – Regions with high precipitation and steep topography shown in yellow where hydropower dams need to provide flood control






The southern Indian state of Kerala, which is bounded by the Western Ghats on the east and the Arabian Sea on the west is a prime example of the need for such a satellite-based reservoir monitoring system (Figure 1). This state of India witnessed its worst ever flooding in history during the month of August 2018 (Joseph et al., 2020). Unexpected and extremely high precipitation forced 35 hydropower dams across the state to be opened simultaneously with little warning leading to an
unprepared flood disaster in the downstream regions. Although the main reason for the flood was attributed to the high rainfall, it was concluded by post flood studies that dam and reservoir operations were also a factor (Mishra et al., 2018). Cross boundary jurisdictional issues further compounded the management of the flooding disaster with the Indian State of Tamil Nadu, a neighbouring state, owning one of the hydropower dams in Kerala but with no framework for state-level bi-lateral cooperation. As a resolution to an inter-state water dispute between Kerala and Tamil Nadu (Thatheyus et al., 2013), control of certain key
dams is under the Tamil Nadu government. Untimely release of water and non-cooperation between the two different state agencies were therefore considered to be a significant contributing factor of the flooding (Sreejith, 2021). Such issues in reservoir management ultimately impacts the lives of thousands and play a significant role in regional water stability. Thus, it is imperative that information regarding the operations of these reservoir be freely accessible, timely and transparent.

Using the Kerala 2018 flood as a case study, this study asks the following research questions. '*How well can we apply a satellite remote sensing and model based framework for near real-time monitoring of the dynamically changing state of hydropower reservoirs in mountainous and high precipitation regions?*' '*With what certainty can such a modelling framework capture what transpired during the flooding event?*' We apply the Reservoir Assessment Tool (version 3.0; RAT 3.0) developed by Minocha at et al. (2023) to pursue this question. RAT 3.0 combines hydrological modelling with high frequency
satellite observations to monitor and predict dynamic state of a reservoir comprising inflow, storage change, surface area, outflow and evaporative losses. The tool has seen application in multiple river basins for operational monitoring such as in Mekong, Tigris Euphrates, Nile, Columbia, and Indus (Das et al., 2022; Hossain et al., 2023). RAT 3.0 is also currently being operationalized for monitoring of the reservoir's dynamic state at about 1600 reservoirs over the world with scripts and data processing methodologies that are made fully public, open-source and scalable. Findings from this study are therefore expected
to inform how RAT 3.0 can be further modified to suit the operational needs of reservoirs in mountainous and high precipitation regions (see yellow regions of Figure 1) that serve the dual role of hydropower generation and flood control.

The paper is organized as follows: Section 2 provides a brief description of the study area of Kerala, over which RAT 3.0 was applied and case study of the 2018 Kerala Floods. Section 3 describes the data, key concepts and the methodology employed. Section 4 discusses results, showcasing how well RAT 3.0 was able to track the events that transpired during the flood. Finally,
Section 5 summarizes the key findings, limitations, and future improvements that can be carried out.



## 2 Study Area: Kerala, India

Kerala is a coastal state in the southwestern region of India, bounded by the Arabian sea to its west and the Western Ghats
Mountain ranges to the east. It has a tropical climate with high temperatures, humidity, and rainfall. Kerala has an extremely
varied topography, ranging from low lying coastal plains to the steep highlands of the Western Ghats. Most of its annual
rainfall of ~2900mm occurs during the Southwest Summer Monsoon from June to September, when warm winds from the
Arabian Sea cause cloud precipitation over the Western Ghats Mountain range (Sudheer et al., 2023) Forty-one of the 44 rain
fed rivers originating from the Western Ghats flow westwards through the state and along with their 30 tributaries, form nine
distinct river basins, namely Achencoil River, Bharathappuzha, Ithikkara, Karuvannur, Karamana, Manimala, Muvattupuzha,
Pallikkal and Kallada, and Periyar (Kumar et al., 2020)A total of 61 dams have been constructed along these rivers, mostly for
energy production and irrigation needs. These dams and accompanied reservoirs cater to ~80% of Kerala's energy needs and
are also the main source for agricultural and domestic freshwater (Sudheer et al., 2019.). Figure 2 shows the location of the
major dams that were considered in this study for understanding the skill of satellite-based framework like RAT 3.0 for
monitoring.

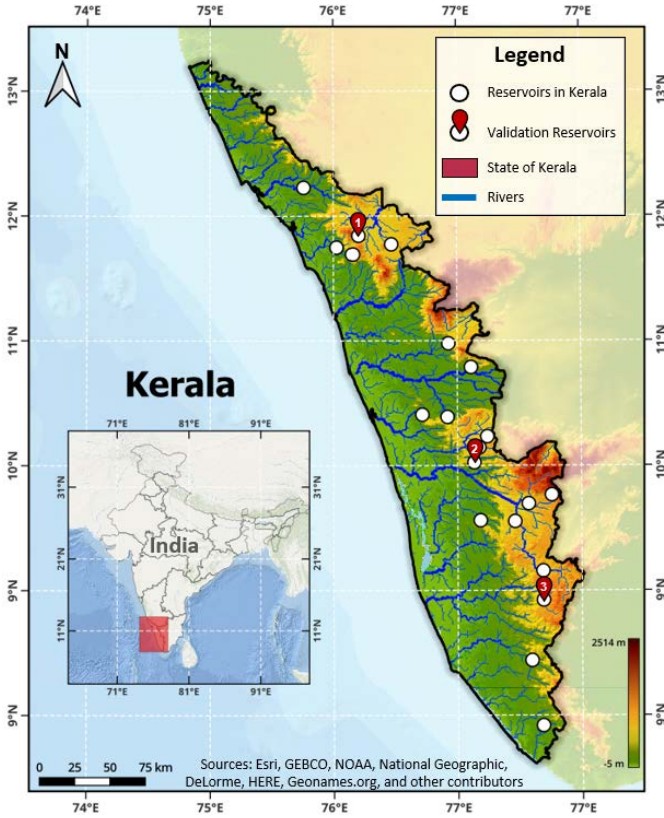

Figure 2: Map of Study area of Kerala, India with location of reservoirs modelled by RAT 3.0. Validation reservoirs for assessing the skill
of RAT 3.0 is shown in red and are as follows: 1. Banasurasagar, 2. Idamalayaar, 3. Kakki





## 2.1 Kerala and the 2018 Flood

In the month of August 2018, Kerala experienced its worst ever flooding in recorded history claiming the lives of over 489
people, inundating thousands of buildings, displacing over 1.4 million people and causing damage in excess of $5 billion
(Pramanick et al., 2022). The flooding was primarily attributed to a propagating low-pressure system from the Bay of Bengal
at the beginning of the month, followed by a monsoon depression. This resulted in the state receiving unusually high amounts
of precipitation (0.6% probability of exceedance) (Sudheer et al., 2019). Thirty-five of the 61 dams in the state were forced to
be opened in a never-before-seen scenario as reservoir levels started nearing dangerous levels, placing all 14 districts of the
state from Thiruvananthapuram to Kasaragod on red alert. For the first time in history, all five overflow gates of the Idukki
dam, India's largest arch dam, was opened at the same time, and for the first time in 26 years, five gates of the Malampuzha
dam were opened (Pramanick et al., 2022). The resultant surge in downstream flow caused the banks of many major rivers
such as the Periyar, Chalakudy, Meenachil, Bharathapuzha, Pampa to overflow, leading to widespread flooding.

The long-term average rainfall for the state of Kerala observed over the period of 1901 – 2017 is about 2400mm with a standard
deviation of 400mm. However, from June 1, 2018 to August 19, 2018, the state received 2346.6mm of rainfall within a span
of 3 months, which is 42% above the expected precipitation for the period (Joseph et al., 2020). Analysis of the return period
of the heavy rains that occurred in the month of August 2018, using Generalised Extreme Value distribution showed that the
1-day, 2-day and 10-day maximum rainfall had a return period of ~75 years, ~200 years and ~50 years, respectively (Mishra
et al., 2018.). The probability of such extreme rainfall in the month of August is very low, at around 0.6% in a year (Sudheer
et al., 2019).

Of the 61 dams in Kerala, 33 dams are controlled by the Kerala State Electricity Board (KSEB), 20 by the Department of
Irrigation, 2 by Water Authority of Kerala, 4 by the Public Works Department of Tamil Nadu and 2 by other entities. Most of
these dams are primarily designed for hydroelectric power generation and irrigation needs and not for flood control. Therefore,
the long-standing operation policy is to maintain the reservoir storage close to Full Reservoir Level (FRL) throughout the
monsoon to ensure maximum power production (Anandalekshmi et al., 2019). During the 2018 floods, in almost all the
reservoirs, the storage level had crossed the spillway crest level by July 26, 2018, and most reservoirs had filled up to over
90% capacity. This caused a dire situation where almost all the flood inflow caused due to the high precipitation had to be
released, essentially removing the role of dams as a regulating structure. The Periyar River basin, which is regulated by 17
dams/barrages, contributing to 80% of the hydroelectric needs of the state, saw the bulk of the flooding (Sudheer et al., 2019).
Heavy criticism was initially placed on the dam management and operations, although further investigational studies presented
a more natural cause for the calamity. Nonetheless, it was made evident in the many post flood analyses that there is an urgent
need for the states dams and reservoirs to shift to a more dynamic operating curve-based operation driven by predicted inflow
scenarios to actively mitigate such flood events in the future (multiple citations). A study on the Kakki reservoir suggested
that, had the reservoir not been at 90% or above capacity, given even just a week to reduce the reservoir storage, a peak outflow
reduction of 50% would have been possible, minimizing the downstream impact of the flood significantly(Ryan et al., 2020).





This case study clearly illustrates the risks associated with utilising hydropower reservoirs in the capacity of flood control. Sudden onset of intense precipitation driven surface runoff could prove too much for such dams to handle, given the nature of their operation. Hence, there is an urgent need for a support system that enables reservoir managers to develop ideas about the possible inflow scenarios. Additionally, there is a need to develop a transparent system for the last mile communication of
flood risk to the most vulnerable population with sufficient lead time.

## 3 Data and Methodology

### 3.1 Data

The RAT 3.0 framework utilises satellite data for estimating reservoir surface area along with meteorological data as forcings for the hydrological model to predict inflow. In-situ inflow and outflow data of the reservoirs are obtained from KSEB for
validation of the RAT 3.0 model outputs. A combination of three satellite sensor datasets namely Landsat-8, Sentinel-2, and Sentinel-1 is used to estimate the surface area of the reservoirs. Microsoft Planetary Computer, which is a cloud-based platform that provides access to large-scale geospatial data, was used to obtain the satellite data. Planetary computer offers a scalable computing solution enabling users to process, analyse and model large volumes of satellite data efficiently, while eliminating local machine storage limitations. Meteorological data comprising of precipitation, minimum and maximum temperature, and
wind speed, utilised as inputs for the Variable Infiltration Capacity (VIC 5.0) hydrological model (Hamman et al., 2018) was obtained from the National Oceanic and Atmospheric Administration (NOAA) and National Centres for Environmental Prediction (NCEP). The properties and source of these datasets are specified in Tables 1 and 2. The Global surface water occurrence dataset (Pekel et al., 2016) is utilised for cloud correction of the optical satellite data.

185          Table 1: Summary of satellite sensors utilised in RAT 3.0 modelling of Kerala

| Sensor | Type | Spatial Resolution | Temporal Resolution (Revisit Period) | Source |
|---|---|---|---|---|
| Landsat-8 MSI | Optical | 30m | 16 days | Microsoft Planetary Computer: Landsat Collection 2 Level-2 |
| Sentinel-1 SAR | Synthetic Aperture Radar (SAR) | 10m | 10 days | Microsoft Planetary Computer: Sentinel 1 Radiometrically Terrain Corrected (RTC) |
| Sentinel-2 A/B MSI | Optical | 10 – 20m | 10 days for a single satellite (~5 days for two satellites) | Microsoft Planetary Computer: Sentinel-2 Level-2A |


Table 2: Summary of the meteorological datasets utilised as forcing inputs in RAT 3.0

| Data | Resolution | Frequency | Source |
|---|---|---|---|
| Temperature – Min and Max | 0.0625° | Daily | NOAA NCEP / Climate Prediction Centre |
| Precipitation | 0.0625° | Daily | NASA: Integrated Multi-Satellite Retrievals for GPM (IMERG – FINAL) |
| Wind Speed – at 10m | 0.0625° | Daily | NOAA NCEP / Climate Prediction Centre |

## 3.2 Methodology

For estimating the inflow and outflow for any given reservoir, RAT 3.0 utilises a simple mass balance approach as illustrated
in Figure 3. The framework consists of the following components:

1. Inflow generation
2. Surface Area estimation
3. Area Elevation curve and Storage change estimation
4. Evaporation estimation
5. Outflow generation.

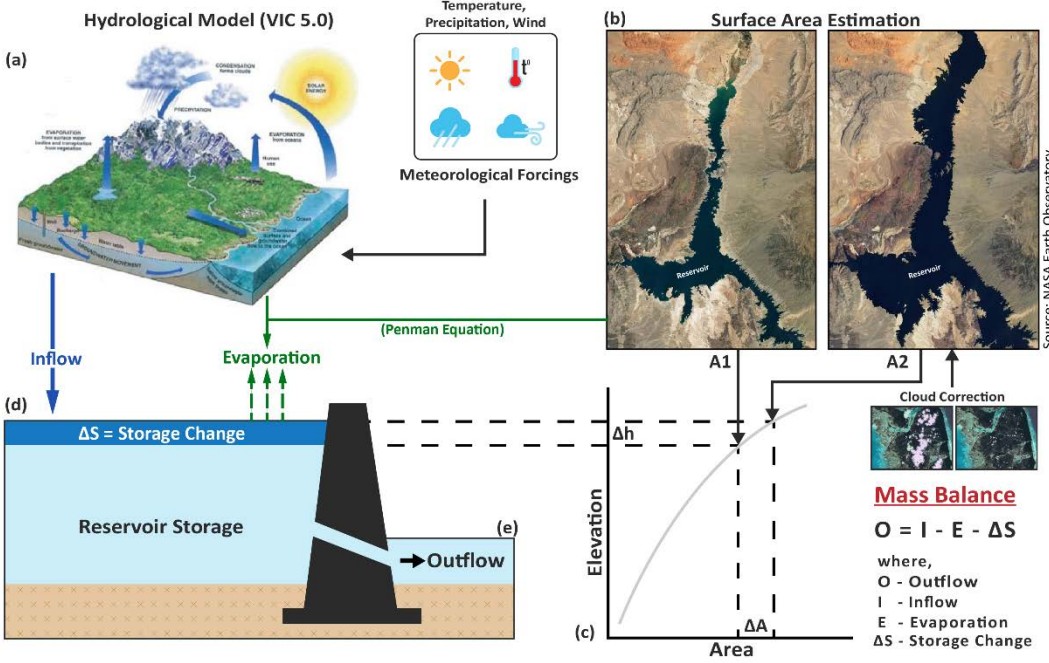

Figure 3: Conceptual model of RAT 3.0. The various components of the framework are as follows: (a) Hydrological model, (b) Surface Area Estimation, (c) Area Elevation curve, (d) Storage change estimation, (e) Outflow estimation.





### 3.2.1 Inflow estimation

RAT 3.0 uses meteorological data as inputs to a hydrological model to generate the inflow. The RAT 3.0 framework is model agnostic and can utilise any hydrological model. In its current implementation, VIC 5 (Hamman et al., 2018), a gridded macro-scale hydrological model that solves full water and energy balances is utilized. Forcing data required as input for VIC 5 is generated using MetSim (Bennett et al., 2020.) a meteorological simulator and forcing disaggregate for hydrologic modelling and climate applications at a spatial resolution of 0.0625°. Runoff produced by VIC 5 is provided as input to the VIC Routing

model (Lohmann et al.,1998) along with the Dominant River Tracing (DRT) based flow direction file (Wu et al., 2011) at the same resolution, to generate the resulting streamflow. The streamflow value is then extracted at the required dam locations to obtain the reservoir inflow.

Gridded precipitation from the IMERG-Final dataset was obtained and clipped to the state of Kerala. In comparison with observed precipitation data as obtained from the Indian Meteorological Department (IMD), the satellite observed precipitation

shows reasonable match with respect to trend and magnitude (Figure 4).

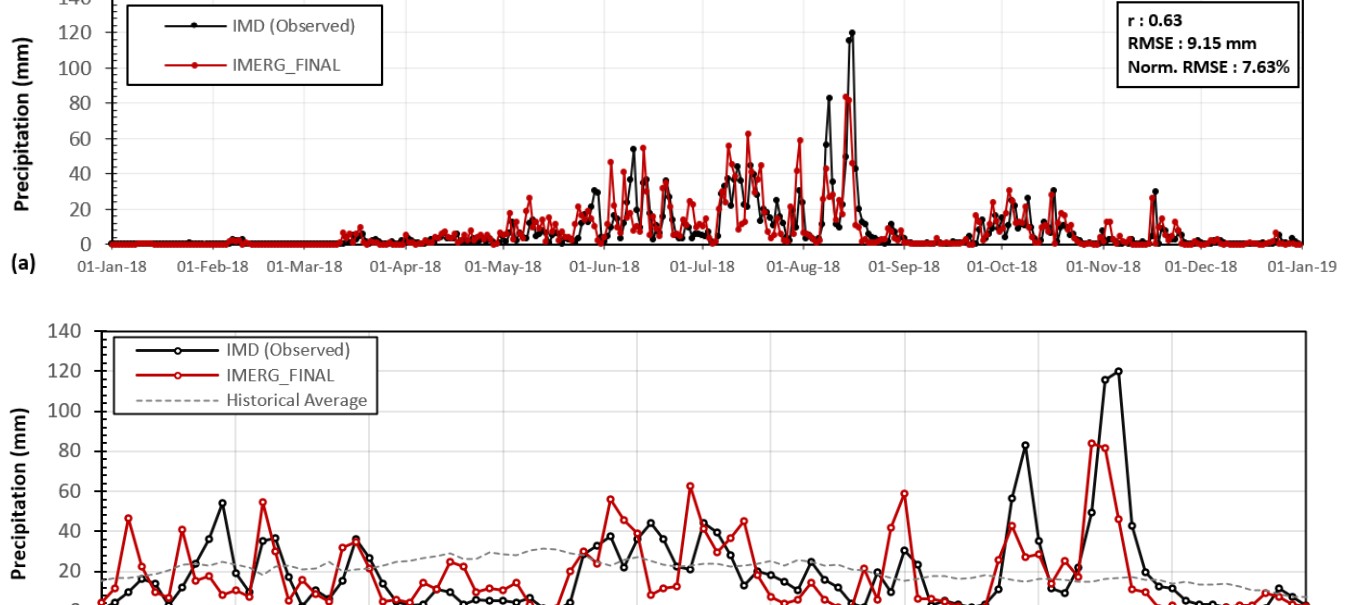

Figure 4: Observed (IMD) and Satellite derived (IMERG) average precipitation for the state of Kerala for (a) 2018, (b) June – August 2018 period of extreme flooding

Hydrological models such as VIC are required to be run for a certain initialization period, known as spin-up time, to attain an equilibrium initial state and produce reliable results thereafter. Spin-up of 2 years from 2015 – 2017 was carried out, the results discarded, and the model state preserved for further model run. The model was then run for the time period of 2017 to 2018 to be used for the current study.  The VIC model and routing model was then manually calibrated with respect to in-situ inflow data obtained from the KSEB for Idamalayaar reservoir. The calibrated model was then run for 20 reservoirs in the state. The



reservoirs were chosen based on their size and in some cases based on political significance such as the Mullaperiyar dam, which is operated by the neighbouring state of Tamil Nadu. The reservoir inflow was validated at two more reservoirs, namely Kakki and Banasurasagar shown in Figure 2 (Table 4).

Comparison of the inflow for Idamalayaar, Kakki, and Banasurasagar (Figure 5 and Table 3) shows that in general RAT3.0 is able capture well the timing of trends in the inflow rise and fall. However, for both Idamalayaar and Kakki, the inflow seems to be underestimated. The difference in magnitude is most noticeable in the case of the inflow peaks.

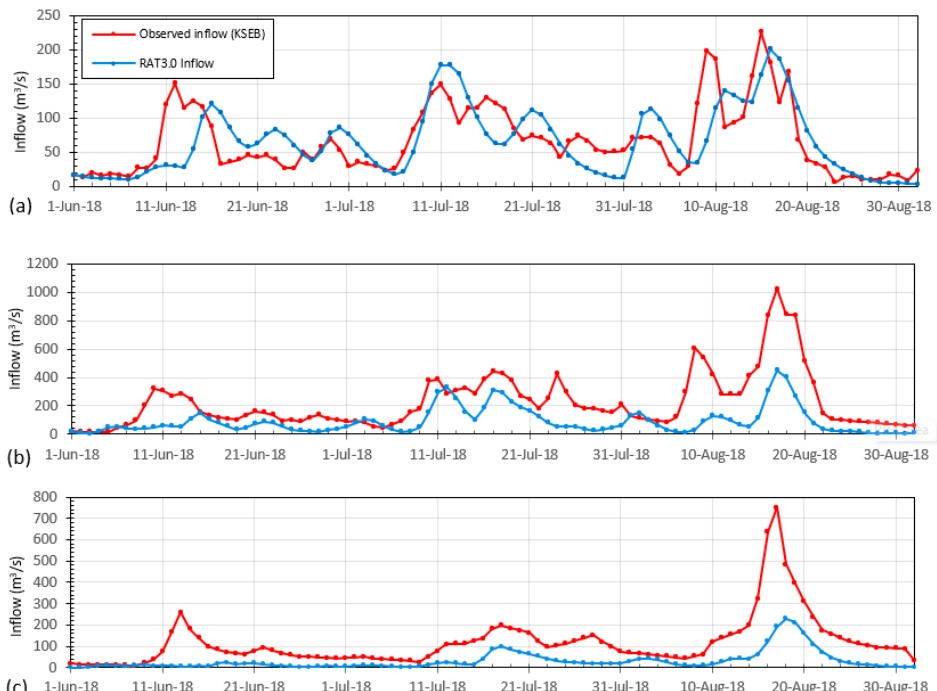

Figure 5: Inflow comparison between RAT 3.0 simulation and in-situ records at reservoirs (a) Idamalayaar, (b) Kakki, and (c) Banasurasagar for period of June to August 2018

Table 3: Statistical comparison of Inflow time series data for the validation reservoirs

| Reservoir | Correlation | Root Mean Squared Error (RMSE) | Normalised RMSE |
| --- | --- | --- | --- |
| Idamalayaar | 0.81 | 135.29 cumecs | 13.30 % |
| Banasurasagar | 0.78 | 31.81 cumecs | 14.14 % |
| Kakki | 0.83 | 83.54 cumecs | 11.19 % |

### 3.2.2 Reservoir Surface Area Estimation

The change in the surface area is required to estimate the storage change and ultimately the outflow from the reservoir. High frequency (1-5 day) satellite data record of the surface area is generated utilising a methodology derived from the Tiered Multi-





Sensor (TMS-OS) approach developed by Das et al. (2022). The TMS-OS approach is summarized here in this section for

reader's benefit. The cloud-based computing capabilities of Microsoft Planetary Computer is utilised for collecting, processing, and analysing the satellite data for a time period of 2017-01-01 to 2018-12-01. Landsat-8 and Sentinel-2 satellite data are chosen for the optical satellite imagery, and Sentinel-1 for the Synthetic Aperture Radar (SAR) dataset. SAR data has the advantage of not being affected by atmospheric effects such as cloud cover and gives a reliable estimate of the surface area trend, while the optical datasets tend to give a more accurate estimate of surface area provided the imagery is cloud free. The

three satellites combined provide a revisit time period ranging from 1-5 days. Surface Area calculation comprises of the following steps as depicted by the flowchart in figure 6:

1. **Water extent mask creation:** The Normalised Difference Water Index (NDWI) and the Modified Normalised Difference Water Index (MNDWI)is used to extract the water area mask from the optical satellite datasets. The Otsu method of automated thresholding (Otsu, 1979) is used to extract the binary water mask from the NDWI and MNDWI images. For

the SAR imagery, VH band from the radiometrically terrain corrected Sentinel-1 imagery is used with a threshold of - 13dB to extract the water extent mask. Due to the steep nature of the topography, high cloud cover, and the extreme spatial and temporal variability of precipitation in the state of Kerala, surface area estimation is always accompanied by a degree of uncertainty. The three different estimates of surface area is used to form the upper and lower bounds for the uncertainty range.

2. **Cloud correction:** High percentage of cloud cover makes surface area estimation erroneous. The Zhao and Gao (2018) method of cloud correction, utilising the historical water occurrence probability dataset (Pekel et al., 2016) was implemented to improve the surface area estimates. The method works by replacing the portion of the water mask overlapped by the cloud mask, with the water mask derived from the Pekel water occurrence dataset, subject to a certain threshold of occurrence (Figure 7 and 8). The reservoir region of interest (ROI) is first buffered to 500m and the surface

water extent mask is obtained through the use of NDWI and MNDWI. The cloud mask is then generated for the scene utilising the quality control bands of the satellite data. Parts of the water extent mask that is covered by the cloud mask is a region of erroneous data. This portion is discarded and replaced by the mask of the Pekel water occurrence dataset (Pekel et al., 2016) for a probability of occurrence greater than 15%.

3. **Filtering:** The optical satellite derived surface area time-series thus obtained can be noisy even after correcting for cloud

cover, especially during instances when a significantly smaller portion of the reservoir is observed by the satellites. This noise issue is mitigated with three additional layers of filtering. The first filter (Filter-1) removes any outliers, defined as those values that differ from the mean surface area by more than 1.5σ, where σ is the standard deviation the surface area time series, over a time period of 30 days. The second filter (Filter-2) computes the bias of the optically derived surface area values from the SAR derived surface area values. If the bias exceeds a threshold defined as 7.5% of the nominal

surface area of the reservoir, then the values are removed. They are replaced with a surface area value such that the bias is now equal to the threshold. The final filtering (Filter-3) performs a SAR trend correction of the optical surface area time series. For every value of surface area, its trend is compared to the weekly trend of the SAR derived surface area time





series. The absolute deviation in the slopes is computed and a threshold of 0.5σ, where σ is the standard deviation of the slope deviations over the past 30 days, is used to remove further unreasonable values. These values are replaced by estimating the missing value using the previous value and the SAR trend as:

$$A'_{t2} = A_{t_1} + SAR.\,trend \times (t_2 - t_1)$$

Where, $t_2$ is the date of the filtered out optically derived surface area value, $t_1$ is the date of the previous unfiltered surface area value, $A_{t_1}$ is the surface area of the previous unfiltered value and $A'_{t2}$ is the missing value of the filtered out optical surface area. The various filtering steps for the Idamalayaar reservoir is illustrated in figure 9.

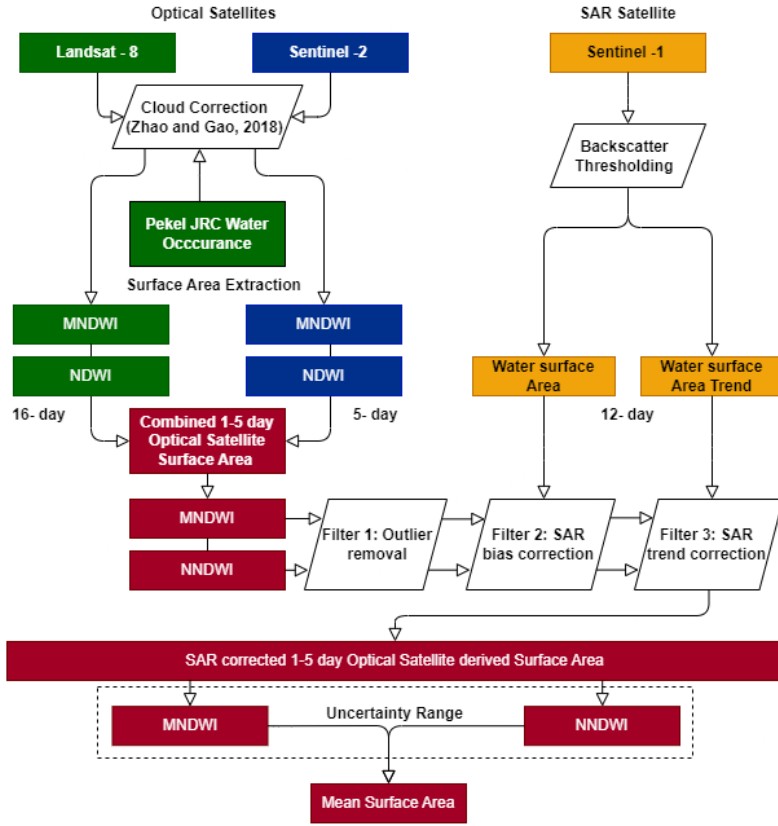

Figure 6: Flow chart describing the surface area estimation process and the various filtering steps.



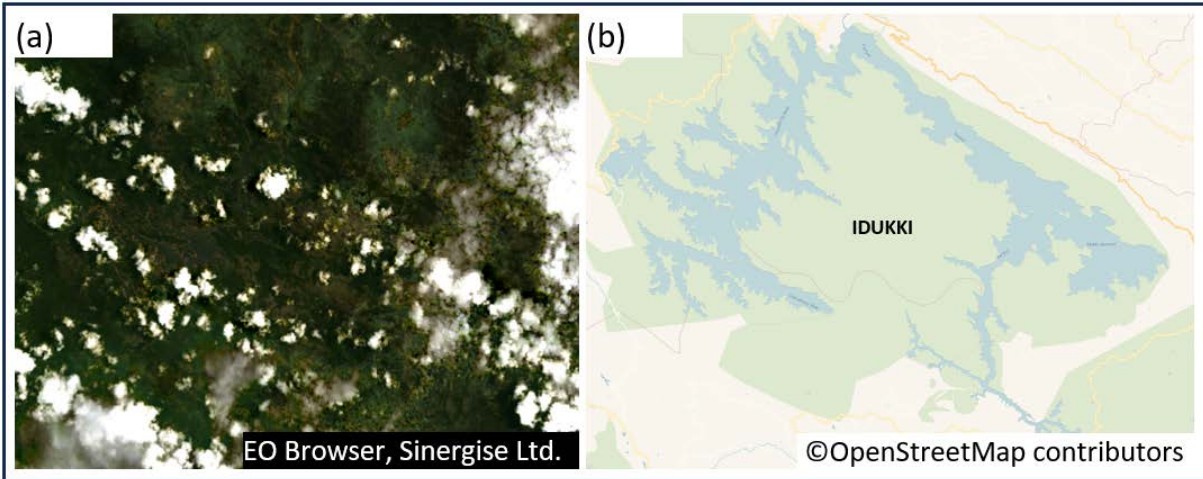

Figure 7: Idukki reservoir located in the Periyar river basin depicted in (a) True color composite alongside (b) reference general purpose map, on 2019-04-03. "© OpenStreetMap contributors 2023. Distributed under the Open Data Commons Open Database License (ODbL) v1.0

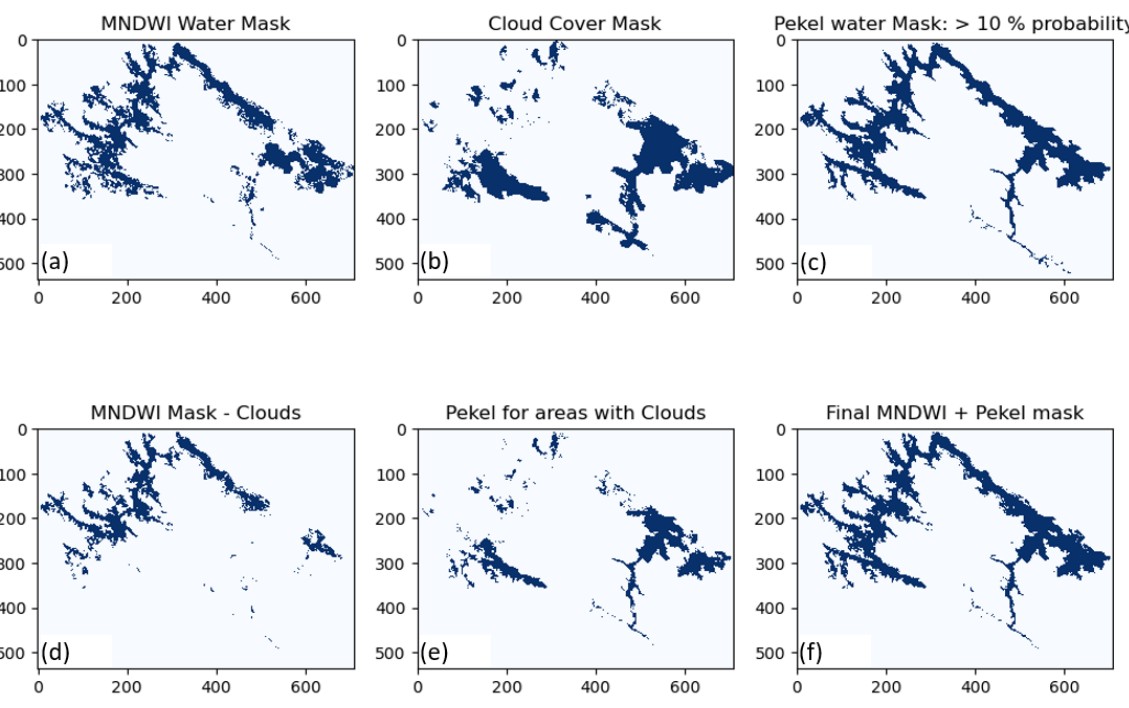

Figure 8: Cloud correction using Pekel water occurrence dataset from Pekel et al., (2016) for Idukki reservoir, Kerala using the method of Zhao and Gao (2018). The inputs required are: (a) Binary water mask, (b) Mask of cloud covered area, (c) Mask of Pekel water occurrence dataset for a selected threshold. The steps involved are: (d) Subtracting cloud mask from water mask, (e) Extracting part of Pekel mask





intersecting with the cloud mask and (f) Merging cloud subtracted water mask and mask containing Pekel data for the cloud covered regions.

The SAR corrected 1–5-day surface area time series for the Idamalayaar reservoir is compared with the observed surface area in figure 10. Overall, the estimated surface area captures the trend of the surface area well while providing higher observational frequency than SAR derived surface area. By combining MNDWI and NDWI based water area estimation, an uncertainty range is provided to the surface area estimate. Two cases of filtering is shown here, a set of reference filters obtained through multiple iterations and a set of highly aggressive filters that causes the Optical surface area time series to more closely mimic

the SAR based surface area. The more aggressive set of filters shows a better correlation with respect to the observed in-situ surface area when compared to the reference filter set (0.94 vs 0.92) but with an increase in the bias (Normalised RMSE of 26.3% vs 22.6% ). Furthermore, due to the very aggressive filtering, the measure of uncertainty is lost as both the MNDWI and NDWI surface areas are smoothened out extremely. The comparison of SAR corrected surface area time-series and observed surface areas for 3 other reservoirs are shown in Appendix Fig A.1


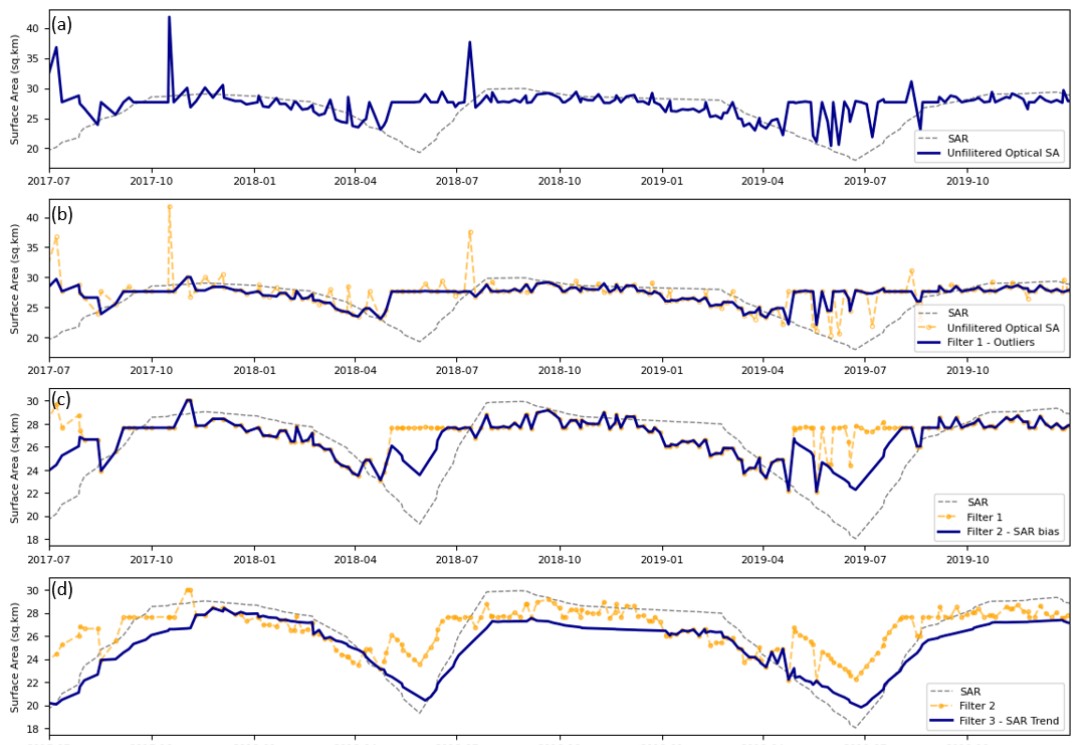

Figure 9: SAR based optical surface area filtering. The various stages are: (a) Unfiltered 1-5 day Optical surface area, (b) Filtering step 1- outlier removal, (c) Filtering step 2 – SAR bias correction, (d) Filtering step 3 – SAR trend correction, which is the final step.




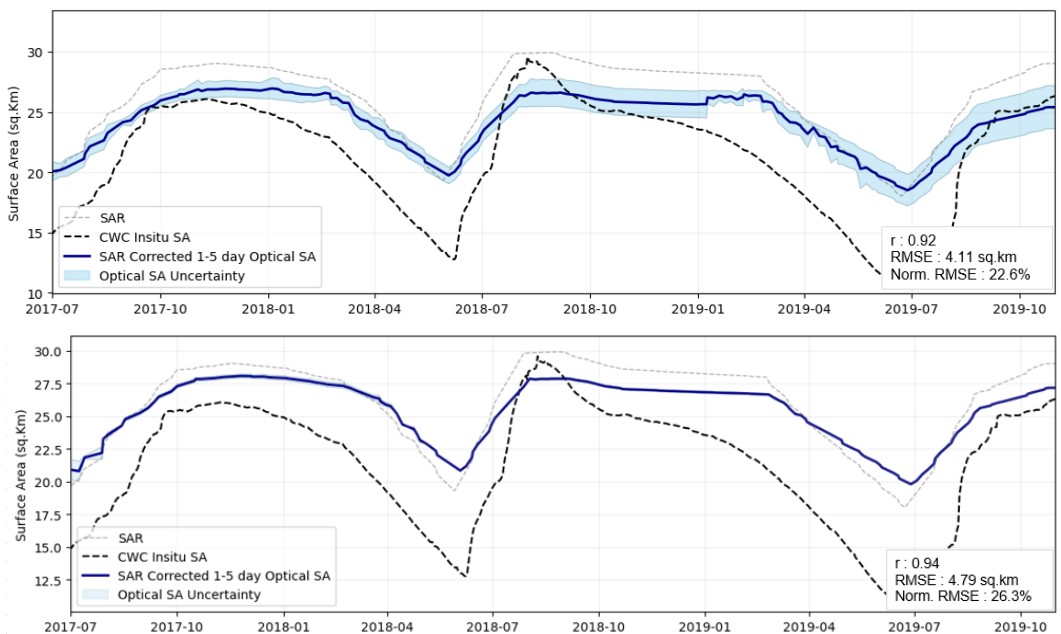

Figure 10: 1-5 day SAR corrected optical surface area with uncertainty range vs Observed surface area from CWC for the Idamalayaar
reservoir for two sets of filters: Upper panel- Reference filter set: filter-1 = 1.5σ, filter-2 = 7.5%, filter-3 = 0.5σ; Lower panel- Aggressive
filter set: filter-1 = 0.5σ, filter-2 = 2.5%, filter-3 = 0.25σ

### 3.2.3 Area – Elevation curve and Storage Change estimation

The change in the surface area of the reservoir can be used along with the area-elevation curve (AEC) of the reservoir to
estimate change in the water level and subsequently the change in the reservoir water storage.  RAT 3.0 calculates the AEC
relationship of each reservoir using the Shuttle Radar Topography Mission (SRTM) 1-Arc Second (30 m resolution) Global
Digital Elevation Model (Earth Resources Observation and Science Centre, 2017). The SRTM mission was a 11-day mission
conducted in February 2000 using a specialised radar instrument aboard a dedicated space shuttle. This restricts the bathymetry
information to that part of the reservoir that was above the reservoir water surface during the mission duration. RAT 3.0
overcomes this limitation by adopting the AEC generation methodology as described in Biswas et al. (2021), which
extrapolates the AEC below the water surface using a fitted power law curve.

The storage change of the reservoir is estimated in RAT 3.0 using the trapezoidal estimation of volume as:

$$\Delta s = \frac{A_{t2} + A_{t1}}{2} \times (h_{t2} - h_{t1})$$

Here, $\Delta s$ is the total volumetric storage change of the reservoir between time instances $t_2$ and $t_1$, $A_{t1}$ and $A_{t2}$ are the
corresponding surface areas of the reservoir, and $h_{t1}$ and $h_{t2}$ are the water levels of the reservoir. The AEC generated





by RAT 3.0 is shown in figure 11. From the curves, we can see that for both Idamalayaar and Kakki, rapid area changes in surface area occur within a small elevation range, indicating that these reservoirs are very mountainous and narrow.

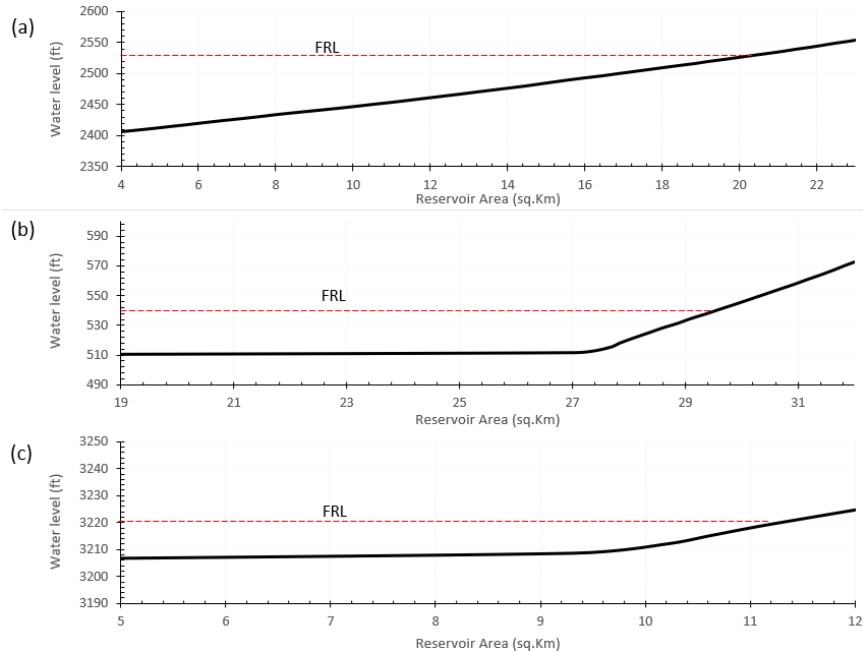

Figure 11: AEC curves generated by RAT for (a) Banasurasagar (b)Idamalayaar and (c) Kakki

Figure 12 shows the storage change estimated by RAT3.0 using the surface area and AEC curves. The storage change estimates are extremely sensitive to minor variations in the surface area estimations, which are reflected by the minor irregularities in the plots.

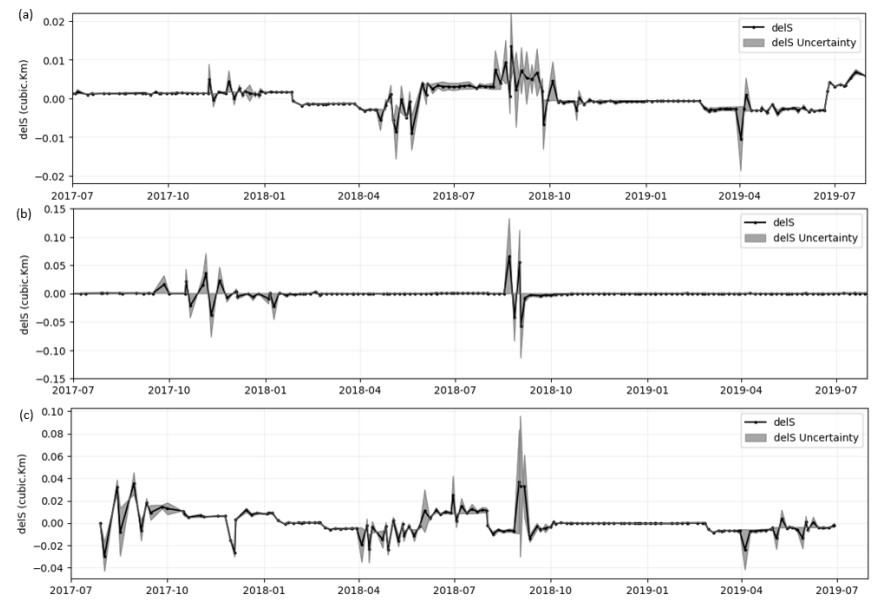





Figure 12: Estimated storage change for (a) Banasurasagar, (b) Idamalayaar and (c) Kakki. The methodological uncertainty in the surface
335            area estimation is carried over to the storage change estimation.

### 3.2.4 Evaporation estimation

The amount of water evaporated from the reservoir varies widely depending on the geographical location of the reservoir and the climatic conditions. RAT 3.0 estimates the evaporation using the Penman equation for free water surface (Penman, 1948, Van Bavel, 1966). The volumetric evaporation $[L^3T^{-1}]$ is generated by multiplying the evaporation with the remotely sensed 340 surface area. Volumetric evaporation time-series for the validation reservoirs as calculated by are given in figure 13. The evaporation rate is very low for all 3 reservoirs with evaporative loss of ~1 m$^3$/s even for the largest reservoir, Idamalayaar.

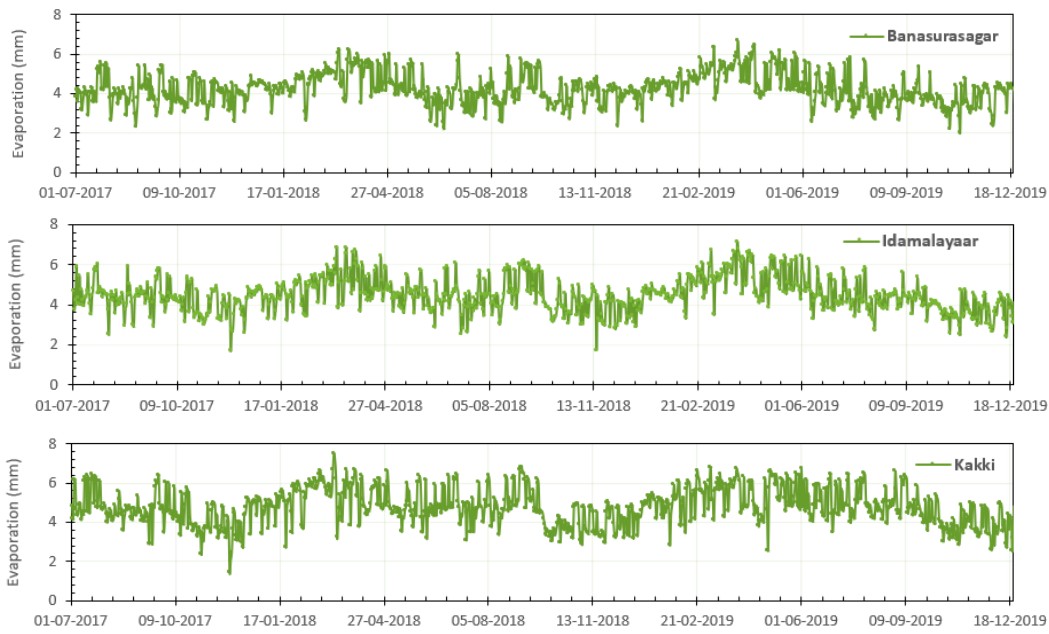


Figure 13: Evaporative loss calculated by RAT for the reservoirs (a) Banasurasagar, (b) Idamalayaar, and (c) Kakki

### 3.2.5 Outflow estimation

The reservoir release or outflow 'O', is estimated using the mass balance approach as:
$$O = I - E - \Delta S$$
Where $I$ represent the reservoir inflow, $E$ is the volumetric evaporation and $\Delta S$ is the storage change of the reservoir. The Outflow thus generated has the same high frequency as the surface area estimates (1-5 days as shown in figure 14. The uncertainty in the estimation of the reservoir water area propagates through to the outflow estimation.





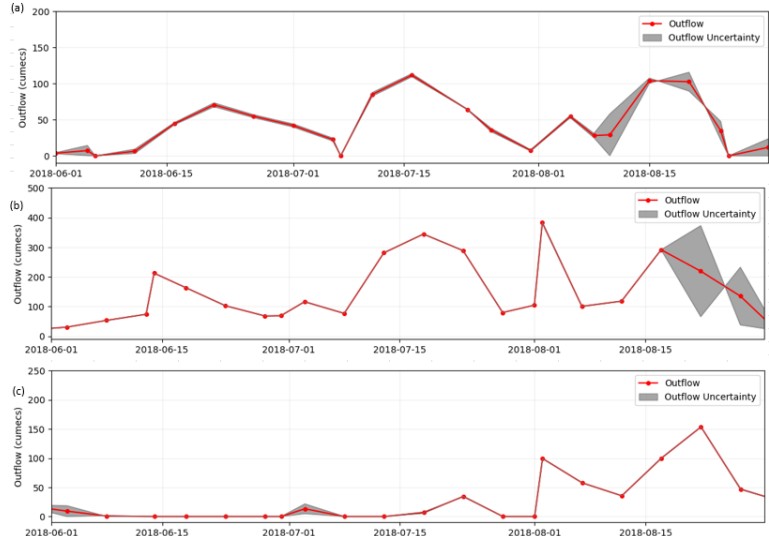

Figure 14: RAT 3 outflows for (a) Banasurasagar, (b) Idamalayaar and (c) Kakki with associated uncertainties. Here, the uncertainty in outflow is propagated from the uncertainty in estimating the reservoir water area and corresponding uncertainty in storage change.

## 4 Results and Discussions

RAT 3.0 was run for a total of 19 dams, spanning 10 basins in Kerala for the period of 2018. The dam and basin details are provided in figure 15 and table 4. To gauge the effective of RAT 3.0 in tracking the events of the 2018 flood, all the reservoirs are analysed with respect to the reservoir water level, inflow, and outflow.

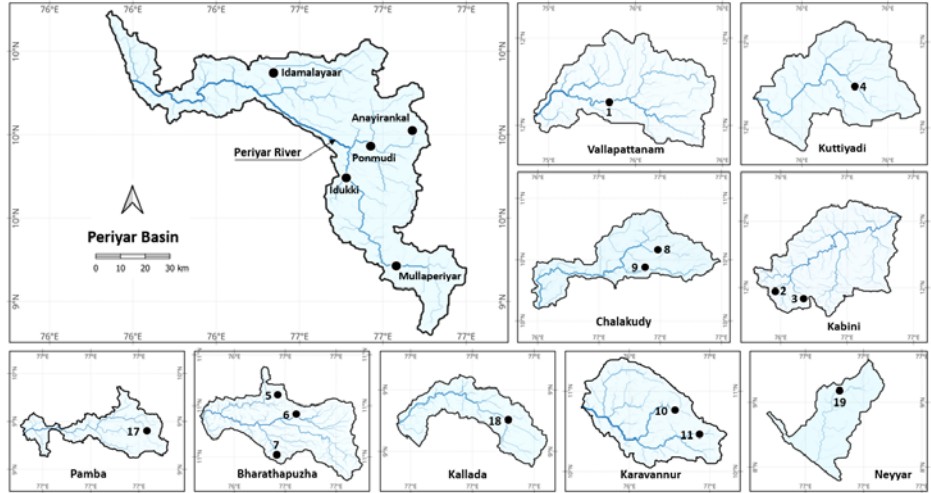

Figure 15: Dams and associated basins analysed using RAT 3.0. Details of numbered dams are provided in Table 5



Table 4: Major reservoirs and corresponding basins in Kerala

| Sl.No | Reservoir | Location | Area | River | Basin |
|---|---|---|---|---|---|
| 1 | Pazhassi | 11.962°, 75.616° | 6.5 km² | Bavali | Vallapattanam |
| 2 | Banasurasagar | 11.671°, 75.96° | 21.2 km² | Karamanathodu | Kabini |
| 3 | Karapuzha | 11.616°, 76.173° | 8.55 km² | Karapuzha | Kabini |
| 4 | Peruvannamuzhi | 11.596°, 75.823° | 16 km² | Kuttiyadi | Kuttiyadi |
| 5 | Kanjirapuzha | 10.986°, 76.541° | 4.65 km² | Kanjirapuzha | Bharathapuzha |
| 6 | Malampuzha | 10.834°, 76.686° | 23.13 km² | Malampuzha | Bharathapuzha |
| 7 | Mangalam | 10.515°, 76.534° | 4 km² | Cherukunnapuzha | Bharathapuzha |
| 8 | Parambikulam | 10.39°, 76.791° | 20 km² | Parambikulam | Chalakudy |
| 9 | Sholayar | 10.319°, 76.738° | 8.7 km² | Chalakudy | Chalakudy |
| 10 | Peechi | 10.529°, 76.37° | 13 km² | Manali | Karavannur |
| 11 | Chimmini | 10.438°, 76.462° | 10 km² | Kurumali | Karavannur |
| 12 | Idamalayaar | 10.222°, 76.706° | 28.3 km² | Idamalayaar | Periyar |
| 13 | Anayirankal | 10.015°, 77.206° | 5 km² | Panniyar | Periyar |
| 14 | Ponmudi | 9.959°, 77.056° | 3 km² | Panniyar | Periyar |
| 15 | Idukki | 9.845°, 76.968° | 60 km² | Periyar | Periyar |
| 16 | Mullaperiyar | 9.528°, 77.148° | 20 km² | Periyar | Periyar |
| 17 | Kakki | 9.341°, 77.15° | 17.6 km² | Pamba | Pamba |
| 18 | Thenmala | 8.954°, 77.074° | 26 km² | Kallada | Kallada |
| 19 | Neyyar | 8.536°, 77.146° | 15 km² | Neyyar | Neyyar |

**4.1 Evaporative Loss**

Evaporative losses as derived using the Penman equation is found to be very low for the reservoirs across Kerala as shown in Table 5. Kakki reservoir had the highest evaporation rate at 7.54 mm/day while Idukki reservoir had the highest total evaporative loss due to its larger surface area. The relatively high humidity throughout Kerala (83% average relative humidity) that causes a state of high-water vapour content in the atmosphere is likely the reason for the lower evaporation rates.

Table 5 : Evaporative losses for Kerala Reservoirs over the period of 2017-2019 as estimated by RAT3.0

| Sl.No | Reservoir | Area (sq. Km) | Evaporation (mm/day) | | Evaporation (cumecs) | |
|---|---|---|---|---|---|---|
| | | | Mean | Max | Mean | Max |
| 1 | Anayirankal | 4 | 4.12 | 6.38 | 0.19 | 0.30 |
| 2 | Banasurasagar | 11.17 | 4.23 | 6.15 | 0.55 | 0.80 |
| 3 | Chimmini | 7.3 | 4.12 | 6.39 | 0.35 | 0.54 |
| 4 | Idamalayaar | 28.3 | 4.52 | 7.05 | 1.48 | 2.31 |
| 5 | Idukki | 50.13 | 4.59 | 6.75 | 2.66 | 3.92 |





| 6 | Kakki | 19.25 | 4.85 | 7.54 | 1.08 | 1.68 |
|---|---|---|---|---|---|---|
| 7 | Kanjirapuzha | 4.51 | 4.27 | 6.74 | 0.22 | 0.35 |
| 8 | Karapuzha | 5.84 | 4.17 | 6.53 | 0.28 | 0.44 |
| 9 | Malampuzha | 18.6 | 4.15 | 6.50 | 0.89 | 1.40 |
| 10 | Mangalam | 2.8 | 4.23 | 6.79 | 0.14 | 0.22 |
| 11 | Mullaperiyar | 21.8 | 4.62 | 6.77 | 1.17 | 1.71 |
| 12 | Neyyar | 8.35 | 4.81 | 7.55 | 0.46 | 0.73 |
| 13 | Parambikulam | 19.45 | 4.57 | 7.54 | 1.03 | 1.70 |
| 14 | Pazhassi | 2.9 | 4.19 | 6.56 | 0.14 | 0.22 |
| 15 | Peechi | 7.98 | 4.16 | 6.63 | 0.38 | 0.61 |
| 16 | Peruvannamuzhi | 8.4 | 4.17 | 6.53 | 0.41 | 0.64 |
| 17 | Ponmudi | 2.2 | 4.64 | 7.08 | 0.12 | 0.18 |
| 18 | Sholayar | 9.1 | 4.68 | 7.61 | 0.49 | 0.80 |
| 19 | Thenmala | 19.8 | 4.22 | 6.86 | 0.97 | 1.57 |

## 4.2 Reservoir Water Area

The SAR corrected 1-5 day reservoir surface area based on the TMS-OS algorithm of Das et al. (2022) was obtained for all 19 reservoirs. The choice of the filtering parameters has significant effect on the estimated surface area. The three filtering parameters control the sensitivity of the optical satellite derived surface area to outliers, deviation from SAR derived surface area, and deviation from the trend of the SAR derived surface area. The effect of change in the three filtering parameters for the Idamalayaar reservoir is shown in Appendix Fig A.2. For reservoirs in Kerala, where the cloud cover is extremely high

(figure 16), aggressive choice of filters was used to ensure that the estimated water area time series is relatively smooth without artificial dips and peaks. The cloud cover over all the reservoirs in Kerala for Landsat-8 and Sentinel-2 overpass is found to be extremely high, with the average cloud cover for the time period of 2017-07-01 to 2019-07-01 being greater than 40% for all the dams. During periods of high precipitation such as the Monsoon period, the cloud cover is over 80%. Hence, satellite-based surface area estimation in such regions is not possible without the use of cloud-piercing SAR imagery.

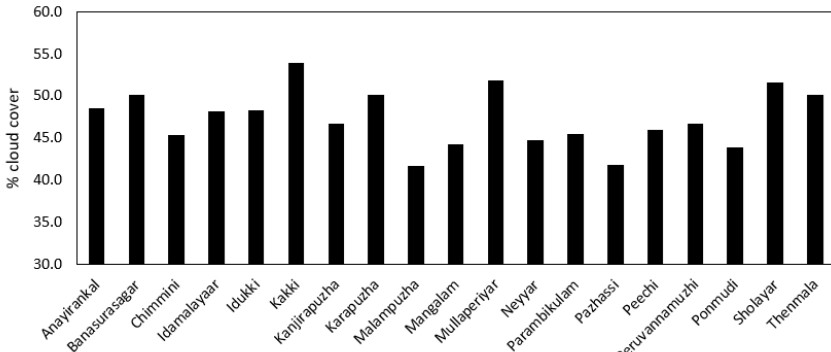


Figure 16: Average cloud cover for Landsat-8 and Sentinel-2 overpass for Kerala reservoirs for the period of 2017-2019





The bulk of the 2018 flooding occurred in the 2nd and 3rd week of August post the sudden the cyclonic depression driven precipitation event beginning on the 8th of August. It was found that the water area of 9 out of 19 reservoirs had already reached 90% of maximum nominal surface area as early as 25th July 2018 (figure 17). By 1st August, 14 out of the 19 reservoirs had reached 90% of the maximum area and by August 8th, 16 out of the 19 reservoirs had crossed 90% of the maximum observed surface area. This translates directly to the reservoirs reaching their full reservoir levels much earlier than anticipated.


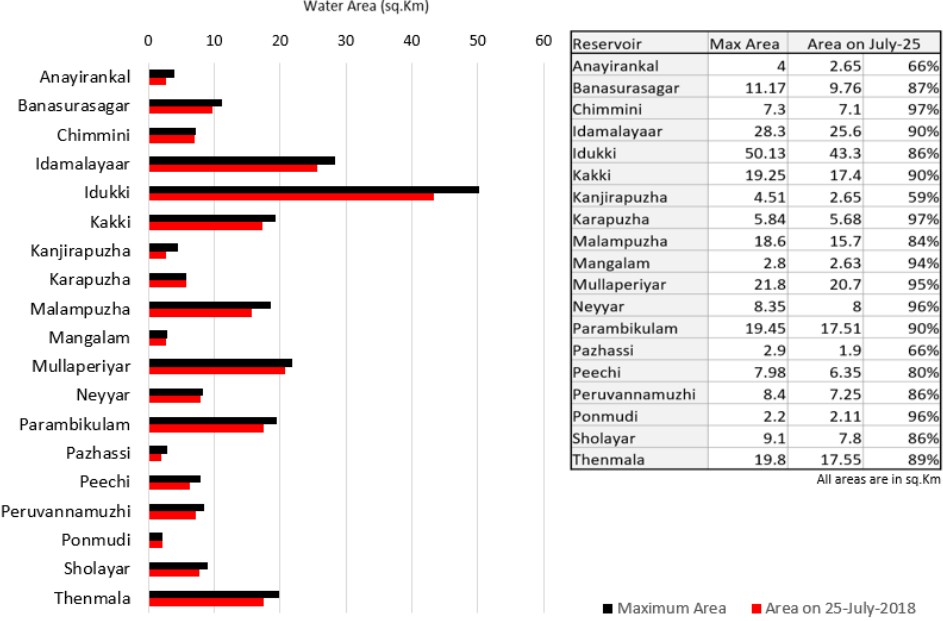

| Reservoir | Max Area | Area on July-25 | |
|---|---|---|---|
| Anayirankal | 4 | 2.65 | 66% |
| Banasurasagar | 11.17 | 9.76 | 87% |
| Chimmini | 7.3 | 7.1 | 97% |
| Idamalayaar | 28.3 | 25.6 | 90% |
| Idukki | 50.13 | 43.3 | 86% |
| Kakki | 19.25 | 17.4 | 90% |
| Kanjirapuzha | 4.51 | 2.65 | 59% |
| Karapuzha | 5.84 | 5.68 | 97% |
| Malampuzha | 18.6 | 15.7 | 84% |
| Mangalam | 2.8 | 2.63 | 94% |
| Mullaperiyar | 21.8 | 20.7 | 95% |
| Neyyar | 8.35 | 8 | 96% |
| Parambikulam | 19.45 | 17.51 | 90% |
| Pazhassi | 2.9 | 1.9 | 66% |
| Peechi | 7.98 | 6.35 | 80% |
| Peruvannamuzhi | 8.4 | 7.25 | 86% |
| Ponmudi | 2.2 | 2.11 | 96% |
| Sholayar | 9.1 | 7.8 | 86% |
| Thenmala | 19.8 | 17.55 | 89% |

All areas are in sq.Km

Figure 17: State of reservoir water area for Kerala dams at the beginning on 25th July 2018. The last column on right shows the actual area

on July 25 as a percentage of maximum area.

It can also be seen from the reservoir water area time series show in Appendix Fig A.3, that for all the reservoirs, flood cushioning is provided as early as June in anticipation of the Monsoon rains. Flood cushioning is the practice of intentionally lowering the reservoir levels, through controlled releases, in order to accommodate an expected increase in inflow. The flood

cushioning that is provided for the Idamalayaar reservoir is illustrated in Figure 18. However, this cushioning was quickly utilised by the continuous precipitation in the months of June and July, leaving insufficient reservoir storage to safely store the additional run-off in the second and third week of August. The continuous Monsoon rainfall prior to the extreme precipitation event had already caused the reservoirs to be near maximum capacity.





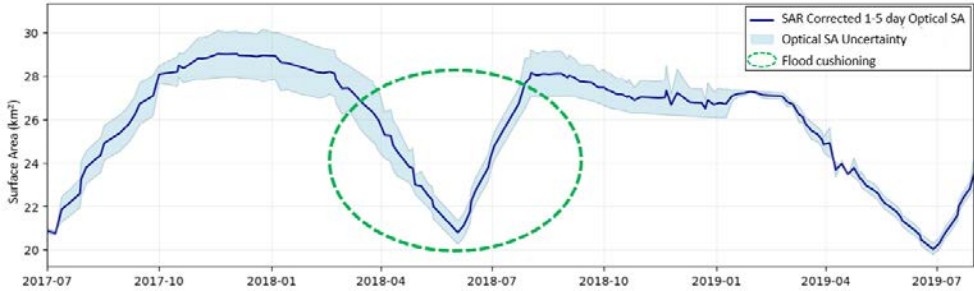

405       Figure 18: Flood cushioning provided in the month of May and June 2018 for the Idamalayaar reservoir. The region marked in green indicates the period where flood cushioning is provided.

## 4.3 Reservoir Inflows and Outflows

Analysis of the reservoir inflows and outflows obtained from RAT3.0 shows that in general, during the period of July to Aug
2018, all 19 reservoirs received large amounts of inflow, significantly exceeding the seasonal average inflow as shown in figure 19. For most reservoirs, it can be seen that the average amount of inflow that was received during the period of July-August in 2018 is 2-7 times more than previous and subsequent years. This unexpectedly high amount of inflow, coupled with the insufficient flood cushioning provided caused the reservoirs to reach the maximum storage quickly and lose any ability to provide flood moderation. Although RAT underestimates the absolute magnitude of the inflows, it is nonetheless very effective
in ascertaining the timing and occurrence of the inflow peaks and consequently in building flood preparedness.

The effectiveness of RAT in tracking the reservoir releases depends on the inflow generated. Figure 20 shows a comparison of outflows obtained by using the RAT generated inflow and outflow obtained using CWC in-situ inflow. Satellite derived surface area is used in both cases. It can be seen that the outflow obtained using RAT inflow is underestimating the outflow, especially during the peak releases. This difference in the two outflows is a measure of the uncertainty with which RAT
estimates the reservoir releases. As it can be seen in the case of the Kakki reservoir, this underestimation of the outflow can lead to a case where the outflow peak is almost missing. This is a direct result of the bias in the inflow estimation. If the estimated inflow peak is small enough, the storage change estimated using the surface area can be large enough to cover up the presence of an outflow peak. Thus, for the effective utilisation of such a satellite-sensor based framework, due diligence has to be taken to ensure that the inflow generated is properly corrected for bias.






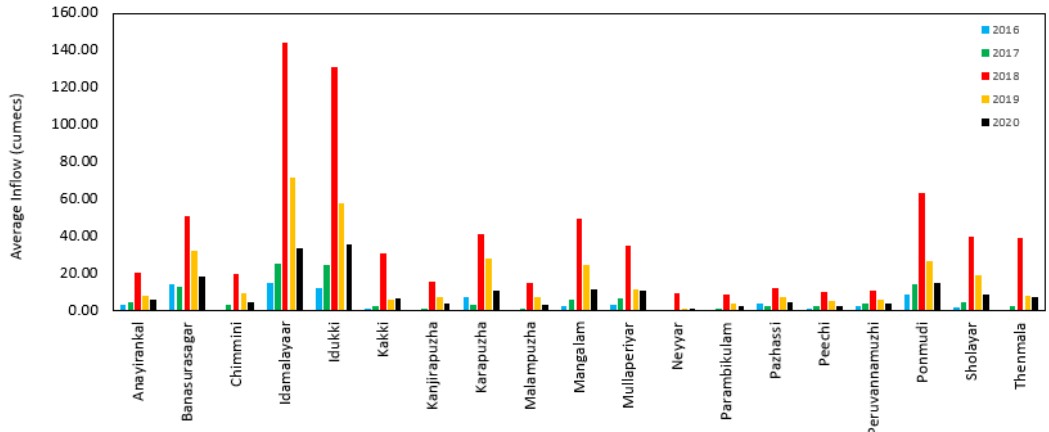

Figure 19: Average reservoir inflow for the period of July-August for 2016, 2017, 2018, 2019, and 2020. The color red is for 2018.

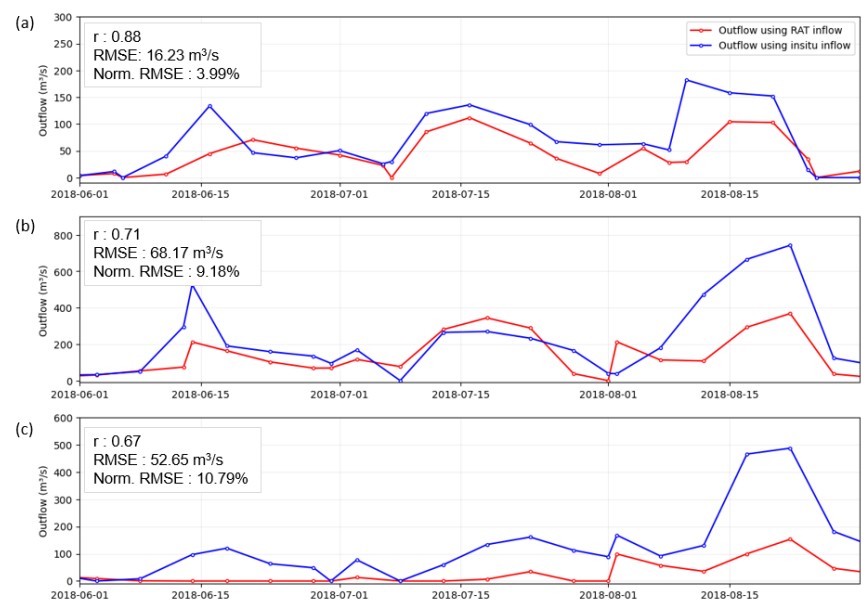

Figure 20: Comparison of outflows obtained from RAT generated inflow (red) and outflow generated using in-situ inflow (blue) for (a)Banasurasagar, (b) Idamalayaar and (c) Kakki

However, the outflow results are still useful in identifying the time of releases given that the absolute magnitude of the outflow can be improved with the use of bias corrected inflow. Figure 21 and 22 shows normalised inflow and outflow of the reservoirs

as heatmaps for the period of July to August 2018. Since the various reservoirs across the state vary in the absolute magnitude of the inflow and outflow, the values are normalised to within the range of 0 to 1 to enable analysis. Min-max normalisation is used for this purpose, which alters the inflow or outflow value as:



$$\bar{x} = \frac{x - x_{min}}{x_{max} - x_{min}}$$

Where, $x$ is a particular value in the inflow or outflow time series, $\bar{x}$ is the normalised value, $x_{max}$ and $x_{min}$ are the maximum

and minimum values of the time series for the selected time period. This causes the maximum inflow or outflow to take the

value of 1 and the minimum to take the value of zero. Areas in the heatmap that are coloured a darker shade of blue or red

indicates the presence of maximum inflow and outflow and those areas that are a lighter shade indicates the absence of inflow

or outflow.

A sudden spike in the inflow is witnessed for 13 out of 19 reservoirs in the second week of July, between the 8[th] and 13[th].

Fifteen reservoirs then continued to receive significant inflow in the third week of July. After a period of normalcy, another

smaller surge in inflow is seen in 12 reservoirs around the first week of August. Finally drastic spikes in the inflows are seen

in all 19 reservoirs from the second week of August with peak inflow occurring around the 17[th] of August 2023. This fully

satellite sensor-based prediction matches with the in-situ reported day of maximum precipitation and inflow (CWC Report,

2018). The reservoir inflows and outflows time series for all reservoirs obtained from RAT are provided in Appendix Fig A.4.

From the outflow heatmap time series plots, it was observed that all 19 dams opened fully in the second and third week of

August, with maximum outflow occurring between 17[th] and 22[nd] of August. This is considered the main cause for the

widespread flooding across the state of Kerala. Subsequently, reduction in both inflows and outflows can be observed in the

fourth week of August corresponding to reduction in the precipitation driven runoffs.

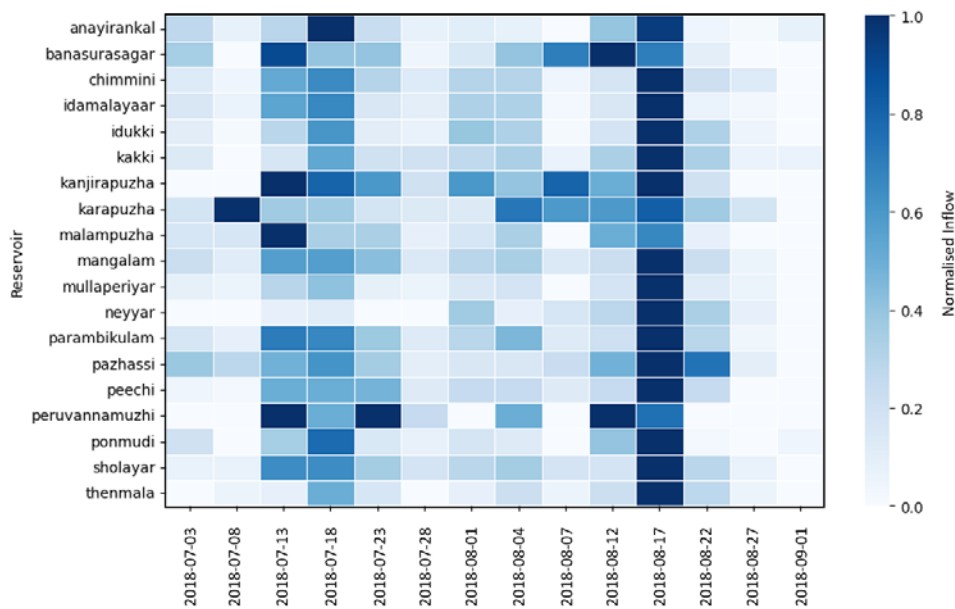


Figure 21: Min-Max normalised reservoir inflow heatmap for Kerala reservoirs for the period of July – August 2018.

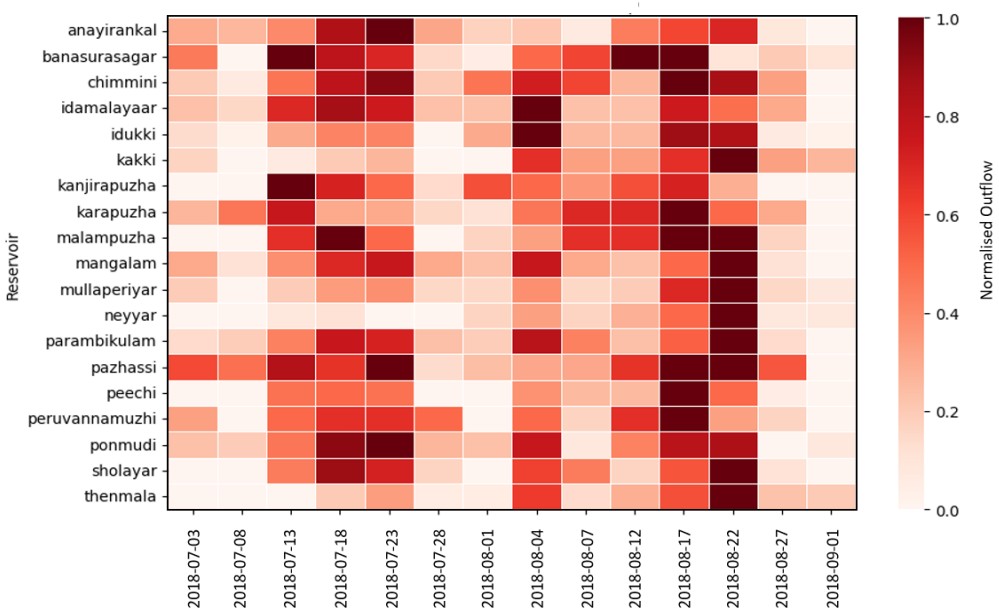

Figure 22: Min-Max normalised reservoir outflow heatmap for Kerala reservoirs for the period of July – August 2018

Thus, a satellite sensor based framework such as RAT does show a strong ability to track the events that transpired during the
extreme weather event, especially with regards to the timings of the peak inflows and releases. However, bias issues in
estimating the absolute magnitude of the inflow and outflows is a limiting factor in utilising the RAT outputs for studies such
as flood height estimation.

## 5 Conclusions

Dams and the associated reservoirs serve as one of the most effective forms of flood control in regions characterised by high
precipitation and steep topography. In such cases, a lack of information of the possible future state of the reservoir leaves water
managers unprepared to tackle unexpectedly high inflows. As can be seen in the case study of the 2018 Kerala Floods, the
standard flood cushioning provided in June to tackle the incoming Monsoon season might become insufficient in the face of
unexpected precipitation events such as one caused due to cyclonic disturbances. It is also impractical to lengthen or delay the

flood cushioning as that would lead to unwanted reduction in the hydropower generation. Thus, the advantages of a reservoir
monitoring framework that is fully space-based and can track the reservoir state becomes apparent. Such platforms enable the
water manager to analyse the reservoir state and inflows to better understand in near real-time how to prepare for a possible
extreme event. A platform such as RAT3.0 which is based purely on satellite derived observations and hydrological modelling
has the added benefit of being truly transparent.





Overall, RAT3.0 was found to be able to track the temporal trends of the reservoir state with good accuracy. The SAR filtered surface area estimates has a frequency of 1-5 days allowing for a near continuous monitoring of the reservoir states for decision making on flood preparedness. The frequency of observations will only keep decreasing with the advent of new satellites such as the Surface Water and Ocean Topography (SWOT; https:// swot.jpl.nasa.gov/). Here, we anticipate SWOT to become a key calibration source for surface water change data for all non-SWOT sensors used in RAT (Sentinel 1, 2, and Landsat 8, 9). The

cloud penetrating properties of SAR satellites such as Sentinel-1 serves as a baseline with which the optical satellites can be corrected. RAT3.0 was able to track the general events of the 2018 floods and was able to identify the peak of the event with certainty. Tracking of the reservoir area is also good and the results are clearly able to identify points in time where the reservoir had reached max water area and the time when the flood cushioning was provided.

However, there are certain aspects of RAT3.0 that needs to be carefully handled to maximise its effectiveness. Inflow into the

reservoirs tends to be underestimated which is likely due to a combination of current limitations of multi-sensor satellite precipitation algorithm and the calibration of hydrologic model (VIC 5.0). From the cases of the Kakki and Idamalayaar reservoirs, it can be seen that the estimated peak inflow is 2-4 times lesser than the observed inflow, although the timing of such a peak was captured very well to trigger warnings promptly for a fully satellite sensor-based monitoring framework. This indicates that the RAT3.0 results are quite useful with respect to the temporal trend of the events but limited in estimating its

actual magnitudes that flood management may require to estimate ensuing flood inundation depths and extent in the downstream. The obtained inflows must be bias-corrected with observed data to effectively capture the magnitude of the inflows. As mentioned earlier, we expect SWOT mission data to mitigate a lot of this uncertainty.

There is also a combined effect of the steep topography of Kerala and the coarseness of the input data used to run the VIC5.0 hydrological model. The resolution of 0.0625° or ~7km may be considered coarse to finely capture the flow directions and

streamflow routing in mountainous regions. The estimated storage change and outflows are also directly dependent on the inflow estimation and on the accuracy of reservoir area trend estimations. Frequent variations in the surface area time series, which is typically seen in optical satellites, may cause erroneous estimation of change in the reservoir storage and corresponding outflows. Thus, aggressive choice of filters must be used to smoothen out the surface area results and minimize false positives or negatives of sudden filling or release. This is mainly an issue caused by high cloud cover. In areas that are

relatively free of clouds, more liberal choice of filtering parameters may be used. Further, inflows into reservoirs that are regulated by releases from upstream reservoirs cannot be tracked using RAT3.0 as of now.

Improvements and additions to the RAT3.0 architecture, as outlined above, are currently under development and we hope to report them in a future study. Such improvements in RAT 3.0 functionality will further increase the utility and usefulness of the tool. Despite the above-mentioned needs for improvements, our study shows that RAT3.0, in its current formulation

(Minocha et al., 2023), is still able to assist reservoir operations during high precipitation events in mountainous topography such as Kerala and make hydropower dam operations potentially more flood safe. As concluded by Ryan et al. (2020), had the water managers been privy to the possible reservoir state of the Kakki dam at least a week in advance of the extreme precipitation event, the reservoir levels could have been brought down in a controlled manner, thereby significantly reducing

the flood peaks. A water manager can utilise RAT3.0 outputs alongside possible inflow scenarios to understand if any dynamic
corrections to the dam operations are required. Running RAT3.0 using forecasted precipitation and judiciously interpreting the
inflow with the current water area and levels given by RAT is another way in which the water manager can quickly get an idea
on the future risk faced by the dam. Incorporating an inflow forecasting mechanism, such as that outlined in the study by Das
et al. (2022), can help achieve this. More involved forecasting techniques and machine learning based inflow forecasting can
help provide the water manager with excellent understanding of future inflow scenarios.

**Code Availability**

The RAT 3.0 software code is available for download from the HELP menu of the RAT 3.0 global web app at
http://www.satellitedams.net Alternately, the code can also be downloaded from https://github.com/UW-SASWE/RAT

**Data Availability**

The in-situ reservoir operations data can be obtained from https://indiawris.gov.in/wris/#/. All other datasets used are freely
accessible. Software documentation on RAT 3.0 can be accessed from http://ratdocs.io User manual is available at:
https://depts.washington.edu/saswe/rat/user_manual/RAT-3.0_User_Manual.pdf

**Author Contribution**

Sarath Suresh: research design, analysis, testing and writing; Faisal Hossain: research design, writing and editing; Sanchit
Minocha and Pritam Das: research design, analysis, editing. All other co-authors: writing and editing

**Competing Interest**

The authors declare that they have no conflict of interest.

**Acknowledgements**

The work on the application of RAT3.0 for Kerala was generously supported by the NASA Applied Science Program through
grants 80NSSC22K0918 (Water Resources) awarded to the second and corresponding author.



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





625



Fig A.4 Reservoir inflow and outflow time series plots with associated uncertainty bands for the outflows obtained from

RAT 3.0