# Peer review of "Satellite-based Tracking of Reservoir Operations for Flood Management during the 2018 Extreme Weather Event in Kerala, India"

_Hydrology and Earth System Sciences, 2023_

## Author Comment (AC1)

**Authors' Response to Reviewer#1**

We sincerely appreciate the reviewer's thoughtful and timely review of our paper. We also thank the editor for securing the review in a timely manner. The key assessment of the reviewer is that our paper *"does not meet the innovation criteria expected for publications in the Hydrology and Earth System Sciences (HESS)"* and therefore they recommend rejection in HESS. While we respect the reviewer's assessment (that they are entitled to), we wholeheartedly reject the notion that our paper does not '*meet the innovation criteria for publication in HESS.*' In our response below we will objectively argue and rebut this assessment. We have italicized reviewer's comments to distinguish them from our responses that are non-italicized.

**Reviewer's General Assessment:** *.....I feel the scope and objective of this manuscript does not fully align with the HESS journal. The focus appears to be more on the specific application case of the RAT 3.0 rather than the broader hydrological and earth system sciences research expected for this venue.*

**Our response:** The first point we should note and clear away for the reviewer is the suitability of our manuscript for HESS given that it was designed  for the HESS Special Issue ***"Representation of water infrastructures in large-scale hydrological and Earth system models."*** One of the editors suggested us to submit our RAT-related work to this HESS special issue. But editor's suggestions aside, our paper is all about '***representation of water infrastructure in large scale hydrological models***' to understand the critical but less-understood problem of *flood preparedness in mountainous regions with high precipitation where hydropower dams operations exacerbate downstream flood risk.* We tackle this problem, global using state of the art data informatics solutions (e.g. cloud computing) and satellite remote sensing. Here the 'water infrastructure' is reservoir/dam that is explicitly accounted for in the modeling/prediction of flood events in fast response, high terrain basins where flood risk management by hydropower dams (that are generally designed to keep full supply level for power generation) are particularly challenging due to its traditional lack of transparency. Around the world, this lack of transparency for the flood management community impacts disproportionately those living downstream who are more vulnerable to dam releases during flood events.

Herein the example we use is the Kerala 2018 floods that the world has documented very well (as we did in the paper) on how mismanaged and uncoordinated the reservoir operations and flood preparedness was due to opaque access to reservoirs' fast changing state in near real-time. We argue that Kerala 2018 floods are not unique – similarly events continue to happen (and have happened) around the world where our study using RAT (optimized for such case) in Kerala are useful. For example, the 2010 and 2022 floods in lower Indus (Pakistan) was exacerbated due to reduced storage and lack of proactive reservoir operations of Upper Indus dams (that optimize hydropower and flood control) in mountainous and high precipitation environments. In a recent paper published in the Journal *Water* by WWF authors Opperman et

al. (2022) (https://www.mdpi.com/2073-4441/14/5/721/pdf), it was reported that *"61% of hydropower dams worldwide will be in river basins with high to extreme risk of water scarcity, floods or both by 2050"* Fig 1 below quoted from the same paper shows exactly how globally relevant (beyond a simple case study) our paper is:

[Figure]

**Fig 1**. Projected increase in flood and drought risk in river basins with existing and planned hydropower projects (Taken from Opperman et al, 2022)

Opperman, Jeffrey J., Rafael R. Camargo, Ariane Laporte-Bisquit, Christiane Zarfl, and Alexis J. Morgan. 2022. "Using the WWF Water Risk Filter to Screen Existing and Projected Hydropower Projects for Climate and Biodiversity Risks" Water 14, no. 5: 721. https://doi.org/10.3390/w14050721

[Figure]

**Fig 2.** (also in the paper) – Regions in yellow show where the findings and lessons learned for RAT 3.0 application over Kerala during 2018 August floods (i.e., our HESS paper) can be applied

around the world where flood risks appearing to be increasing due to the combination of climate change, energy production requirements and land use change.

To the best of our knowledge, *such work described in our paper on representing water infrastructure (hydropower dams/reservoirs) in a hydrological model for highly mountainous, high precipitating environments with a high degree of hydropower generation, is fundamentally missing in literature.* So we believe our work is innovative and a key contribution to the body of knowledge because we have identified scalable or generalizable methods for dealing with the flood preparedness issue in similar environments around the world (see yellow regions in Fig 2 below). Some of the key scalable findings are (which we plan to make clearer in our revised manuscript):

1) In mountainous, coastal and fast response basins, RAT3.0 was found to be able to track the temporal trends of the reservoir state with good accuracy. However, tracking reservoir storage change at the highest frequency and accuracy is more important for such cases. Herein, we argue that the SWOT mission along with the suite of nadir altimeters to track reservoir elevations will play a positive role.

2) Given that RAT is model agnostic, mountainous regions require improved and better calibrated hydrologic models or reservoir inflow. In particular, the streamflow routing scheme requires attention as the area draining into the very upstream reservoirs is quite small in such highly mountainous basins where the dams are often at the edge of the boundary. This is where strong engagement from local partnering agencies to improve the calibration of the model (VIC in our case of RAT 3.0) is critical. Fortunately for Kerala, we are already engaged with Kerala Centre for Water Resources Development and Management (CWRDM) and Kerala State Electricity Board (KSEB) who have agreed to help address this issue.

2) Because of perennial high cloud cover in such regions around the world with hydropower dams (see Fig 2 yellow regions), microwave/radar-based satellite sensors are more critical and play a central role in tracking reservoir state. We recommend that SWOT KaRIN sensor with as many radar altimeters (Sentinel 3A, 3B, 6, SWOT altimeter) be used for tracking reservoir storage change as accurately and frequently as possible for such regions around the world identified in yellow in Figure 2.

The reviewer should note that our paper was framed to answer the following unanswered research questions looking beyond a case study in Kerala, India: "*How well can we apply a satellite remote sensing and model-based framework for near real-time monitoring of the dynamically changing state of hydropower reservoirs in mountainous and high precipitation regions?' 'With what certainty can such a modelling framework capture what transpired during the flooding event?"* We believe our paper tries to answer these questions objectively, truthfully, and transparently (mentioning also the limitations and making our work reproducible via GitHub repository).

In summary, we therefore argue that our work is more than just a case study of RAT for Kerala Although we had articulated novelty of the work in lines 28-115, we plan to integrate a concise summary of the above (4-5 lines) in the revised introduction section of the paper to make it clearer how our study is more than just a case study and how it makes a contribution to the body of knowledge.

**POINT BY POINT RESPONSE TO THE REVIEWER**

*Reviewer Point 1: The work, in its essence, appears to be more of a test case for the Reservoir Assessment Tool (RAT 3.0) developed by the team, and the related manuscript (Minocha et al.) is currently under review for the journal Geoscientific Model Development. A large portion of the methodology section introduces and demonstrates the effectiveness of RAT 3.0, showing a lot of overlaps with the RAT 3.0 paper.*

**Our response:** The reviewer is correct that a portion of the paper is similar in content to the GMD paper that describes the RAT 3.0 architecture and software fully for users, modelers and developers. However, we do not believe our paper is just a test case (please see above our rebuttal why). Also the overlap with GMD paper is intentional so that the reader can find most of the basic details of how RAT works in the HESS paper without going too much into details. The reviewer should note that we provide the basic details in concept only as most of the RAT 3.0 implementation (the TMS-OS algorithm for tracking reservoir storage change – see Das et al. 2022) had to be re-implemented using the Microsoft Planetary Computer API for python as Google Earth Engine did not have all the satellite datasets for the Kerala 2018 floods. Because we believe in open science and FAIR principle (Findable, Accessible, Inclusive and Reproducible), we not only provided the basic description of RAT components, we also provide the GitHub repository for RAT-Kerala so that the paper is self-contained. We therefore disagree that there is a lot of overlap. In fact, the GMD paper talks about software architecture and key innovations from RAT 2.0 and RAT1.0. In our HESS paper, we mainly talk about how RAT works conceptually in terms of its major physical modeling components (hydrology, reservoir and outflow estimation) in a mountainous and high precipitation environment. Our paper also serves the important purpose of real-world application and validation of the RAT framework which is not present in the GMD paper. In response to the same reviewer (comment#6), we now provide most of the RAT3.0 description as an Appendix, which now reduces the 'overlap' this reviewer is referring to.

*Reviewer Point 2: The present manuscript does not seem to offer novel insights and knowledge, or models and tools. Instead, it primarily showcases a case study using the RAT 3.0 to a specific region. Although valuable in its right, it does not meet the innovation criteria expected for publications in the Hydrology and Earth System Sciences (HESS).*

**Our response:** Please refer to our earlier response on previous pages to the reviewer's general assessment. Given that the paper was designed and written for the HESS Special issue with an exclusive focus on dams in highly mountainous, high precipitation regions around the world (to

show that RAT can work with the caveats), we argue again that our paper DOES offer novel insights and knowledge (we have summarized three key novel insights earlier for the reviewer).

**Reviewer Point 3:** *In light of the aforementioned reasons, I recommend that the manuscript be rejected for publication in the HESS journal. Nevertheless, I would like to emphasize that the value of applying the RAT 3.0 is evident, and it may find a more suitable journal that focuses on case studies related to hydrology and reservoir management.*

**Our response:**  We respect the reviewer's recommendation, but we also respectfully disagree with their decision objectively (please see our preceding rebuttal). Further, we should point out to the reviewer that the HESS special issue is a 'suitable journal' that the reviewer suggests as the special issue invites work on water infrastructure in modeling systems. We are also happy to report to the reviewer that two real-world water agencies (CWRDM  and KSEB) have given our RAT 3.0 for Kerala a strong endorsement for its use in their decision-making after we showcased in Sept 2023 the final tool and how it could be operationally useful for the agencies' mission. We would like to point out that our work submitted to HESS special issue is not just an academic exercise – it was driven by real-world and urgent need to co-develop user-ready solution from research (see http://depts.washington.edu/saswe/kerala) for managing flood risk that is exacerbated by water infrastructures. In the revised manuscript, we will provide evidence of this real-world engagement and the user uptake to solve the problem of flood preparedness in an mountainous regions.

**Reviewer Point 4:** *Figure 5 shows a noticeable discrepancy between the RAT-simulated streamflow and observed values, particularly during peak streamflow. Given the significance of streamflow (or inflow to the reservoir) for flood management, any such deviation could raise questions about the reliability of RAT. It is essential to delve deeper into this issue. Here are some suggestions for further investigation that might be helpful:*

*Could you identify the source of this deviation? Is it stemming from the VIC runoff simulation, the routing process, or another aspect?*

**Our response:**

While we acknowledge that there is discrepancy between the RAT-simulated inflow and the observed values, we would like to stress that the discrepancy is for the most part a bias issue. The modeled inflow matches well with respect to the trends (Table 3 in manuscript ) giving us a good qualitative estimate as to the presence of peaks or dips in the inflow. We should also note that the reservoir in the first panel in Figure 5 shows good match in terms of the magnitude of the peaks also.  We believe the bias in inflow estimates as shown in Figure 5 is due to the routing of the VIC modelled inflow. The key limitation is the coarseness of the grid size used (0.0625° ~ 6km) coupled with the mountainous topography of Kerala. Moreover, many of the dams are at the edges of their respective river basins as shown in Fig 3. This causes issues with flow routing where the steep topography sometimes causes flows from some model grid cells

to be not routed correctly to the dam locations. In total 15 out of the 19 dams studied lie extremely close to the basin boundary with very little drainage area.

[Figure]

Fig 3. (also part of Figure 15 in manuscript)  Some of the river basins have dams that are located within 1 grid cell of basin boundary.

We plan to address these bias issues by calibrating the modelled inflow with in-situ values. As mentioned earlier, we have already engaged with two agencies from Kerala, CWRDM and KSEB to obtain the data required for this (to be summarized in appendix of revised manuscript).

*Have you considered quantifying this simulation discrepancy using a stochastic approach, such that the uncertainty can be quantified?*

**Our response:** No. Since RAT is model agnostic, we believe a better approach is to improve calibration of the model, swap the model or use alternative sources of inflow (such as from KSEB and CWRDM). We should note again that the discrepancy is only in magnitude but not in the timing of the rise and timing of the peak inflow, which are as important or more important parameters for flood preparedness. And oftentimes, the magnitude is also captured well (see Figure 5 uppermost panel).

**Reviewer Point 5:** *I have explored the RAT 3.0 (https://depts.washington.edu/saswe/rat/) and appreciated the user-friendly interface. However, under the "MONITOR" or "ANALYZE" tabs, there's a noticeable limitation in the number of river basins available. Additionally, the data for certain reservoirs stops around mid-2022, suggesting a lack of real-time updates. While I understand that there might be computational constraints preventing real-time VIC execution, readers might question the claim of global coverage and real-time monitoring made in the manuscript. It might be prudent to moderate such claims in the manuscript and acknowledge existing limitations.*

**Our response:** Thank you for the comment. Please note the paper is not about the RAT's global portal or the software architecture (which is described in more detail in the GMD paper). We just happened to mention the links to where reader can find all the information to reproduce our finds in the spirit of Open Science and FAIR principles. We appreciate nevertheless the reviewer checking out the www.satellitedams.net site (or http://depts.washington.edu/saswe/rat).

We should correct the reviewer that on the global portal for RAT 3.0 they can actually see the operational state of reservoirs (as recent as Oct 2023 as of writing this response) for Tigris-Euphrates, Kerala, Indus and Mekong. For these regions, RAT is running as a cron job. Only the Texas one hasn't been operationalized as a cron job and we guess that is what the reviewer happened to check at the time of their website visit. Also, the reviewer should note that RAT 1.0 was actually set up over 1600+ dams at http://depts.washington.edu/saswe/rat_beta and that a lot of our RAT is actually running in dedicated systems' front end (such as one for Kerala at http://depts.washington.edu/saswe/kerala Mekong http://depts.washington.edu/saswe/mekong, Nile at http://depts.washington.edu/saswe/nibras (to be restarted as cron job).

The issue is not with CPU limitations per se. RAT 3.0 actually uses several memory and CPU efficient techniques such as hot start for hydrologic model (to avoid spin up each time step during cron jobs), parallelization for input data preparation and models runs etc. We haven't rendered all of the dams yet (we plan to host 7000+ by late 2024) on www.satellitedams.net yet as we are progressively and methodically first completing the stand alone systems for our engaged end users to maximize real-world impact rather than just displaying RAT a fancy site that no one uses.

Finally, we are not sure what 'claims' the reviewer is asking us to moderate without pointing us to the line numbers. We have built RAT 3.0 for the global community to allow empowerment for users/developers to build the tools to model water infrastructure without needing our help. This is in the spirit of Open Science and FAIR and the evidence can been in the recent RAT3.0 downloads by worldwide uses. Our goal in RAT 3.0 software is to lower the barrier of entry and make it easier for anyone anywhere to set it up as a software prior to necessary calibration and improvement that we believe is the responsibility of the user.

The RAT python package hosted in the conda-forge repository has seen a total of 912 downloads so far (since April, 2023) as shown in Fig.4, showing a healthy uptake of the tool. Of these around 40%-50% downloads are from outside the University of Washington. The download count can be viewed using the following url: https://github.com/UW-SASWE/rat-feedstock.

**Current release info 🔗**

[Figure]

| Name | Downloads | Version | Platforms |
|---|---|---|---|
| recipe rat | downloads 912 | conda-forge v3.0.1 | platform noarch |

Fig4. Current RAT downloads from the conda-forge python repository as of 10-12-2023.

**Reviewer Point 6:** *The manuscript devotes a significant portion, from line 171 to line 357, to detailing RAT 3.0 and its effectiveness. This extensive coverage not only seems redundant with the RAT 3.0 paper but could also detract from the main focus. I would suggest relocating the majority of these details to the supplementary materials, providing a concise overview of the primary features and functionalities of the assessment tool within the main text.*

**Our response:** This is a good suggestion. Thank you! We wholeheartedly agree. We therefore have relocated most of the description of RAT to the Appendix section of the manuscript and have only provided the basic overview in the main body along with the revised implementation of the TMSOS algorithm using Microsoft Planetary Computer.

We have also articulated the innovative aspects and key findings in an improved manner in the manuscript so as to highlight that the work is not only a case study but is rather about applying RAT to answer the broader question of its effectiveness in tracking flood events in high precipitation and mountainous regions. The current changes made have been shown in Appendix Fig1 here.

We will combine these change with revisions from other reviewers. However, we do want to note, as already mentioned earlier, it is important to provide sufficient details on RAT 3.0 (independent of the GMD paper) so that the paper stands on its own and readers can reproduce our findings independently.

**APPENDIX**

**1.0 Changes made to original manuscript based on Reviewer 1 feedback**

Based on comments from reviewer 1, we have relocated most of the finer details in the methodology section to the Appendix portion of the manuscript.

[Figure]

Appendix Fig1. Relocating detailed methodology of RAT3.0 components to the Appendix section of the manuscript.

We have also highlighted the innovative aspects and key findings of this paper in an improved manner in the 'Conclusion' section of the manuscript.

[Figure]

Appendix Fig2. Highlighting the innovative aspects and other key findings of the study in an improved manner in the original manuscript.

**2.0 Engagement with CWRDM and KSEB**

We have been engaging with two agencies from the state of Kerala to further improve RAT and to obtain the necessary data for validation. The tool was presented to engineers from these organizations and their feedback and suggestions were received.

The interaction with them and the feedbacks received have been summarized below:

1. CWRDM (on 21-09-2023, by Vivek Balakrishnan, Scientist at KSCSTE – CWRDM)
   *"Overall, RAT (on [http://depts.washington.edu/saswe/kerala](http://depts.washington.edu/saswe/kerala)) shows promise as a transparent and public data-sharing platform with regard to the Kerala reservoirs. It offers a one-stop platform for the tracking of the various reservoirs across Kerala using satellite based observations."*

   Feedbacks:
   - RAT results, although captures the trend of events well, are in need of bias correction with respect to the absolute magnitudes.
   - Long terms inflow time series may be validated with observed data to better understand VIC modelling efficacy.
   - AEC generation method using SRTM was previously tested by CWRDM and was found to be lacking in accuracy. In-situ AEC observations will be more suitable.
   - The potential of SWOT as a means to improve water level and storage change estimates were noted.
   - The possibility of adding RAT-Kerala to the CWRDM website was discussed. Potential of running RAT natively on CWRDM machines was also discussed as a possible option.
   - CWRDM has also developed a gridded precipitation forecasting system. This may be clubbed along with RAT in the future for forecasted inflow and outflow scenario predictions.

2. KSEB (on 25-09-2023, by Dr. Biju P.N, Deputy Chief Engineer KSEB)
   *"RAT displays (on [http://depts.washington.edu/saswe/kerala](http://depts.washington.edu/saswe/kerala)) good potential as a platform to monitor various hydro-electric dams across the state of Kerala. It can aid in flood preparedness in the future and help improve public access to reservoir data."*

   Feedbacks:
   - Necessary data for more calibration or validation can be provided and has to go through Kerala state government channels in the form of official communications.
   - Evaporation estimates using the Penman method may be validated with observed values.

- The reservoir of Idukki is to be validated with observed data as it is the largest hydropower dam in the state.
- Temporal frequency of the observations may be improved from 1-5 day for a more effective flood monitoring system.

---

## Author Comment (AC2)

**Authors' Response to Reviewer#2**

We thank the reviewer for their timely review of our paper. We also thank the editor for securing the review in a timely manner. The overall assessment of the reviewer on the paper is that *"The current work and methodology seem interesting and modern and could offer some additional tools for water resources and flood risk management, although, I think it fits better in a journal more dedicated to remote-sensing."*

The reviewer has also raised some key questions and issues with the manuscript. In our response below, we will objectively address the various points raised by the reviewer. We have italicized reviewer's comments to distinguish them from our responses that are non-italicized.

**Reviewer's general Assessment:** *"The current work and methodology seem interesting and modern and could offer some additional tools for water resources and flood risk management, although, I think it fits better in a journal more dedicated to remote-sensing."*

**Our response**:

We sincerely thank the reviewer for acknowledging the potential of the work with respect to flood risk management.

While it is true that remote-sensing is a key part of our paper, it is to be noted that the work offers a unique solution and physical insights (see our response to reviewer #1 and conclusion of the revised paper) to the less-understood issue of tracking reservoirs in high precipitation and mountainous regions. Here we have explored flood moderation, through the marriage of hydrological modeling and remote-sensing powered by cloud-computing, for mountainous and high precipitation regions such as Kerala that current literature has not yet addressed.

Also, the reviewer should note that our manuscript was designed for the HESS Special Issue **"Representation of water infrastructures in large-scale hydrological and Earth system models"** wherein one of the editors suggested us to submit our RAT-related work. But editor's suggestions aside, we want to make it clear to the reviewer that our paper is all about '*representation of water infrastructure in large scale hydrological models*' using state of the art data informatics solutions (e.g., cloud computing), hydrological modelling and satellite remote sensing. Here the 'water infrastructure' is reservoir/dam that is explicitly accounted for in the modeling/prediction of flood events in fast response mountainous basins where flood risk management by hydropower dams (that are generally designed to keep full supply level for power generation) are particularly challenging due to its traditional lack of transparency. This lack of transparency for the flood management community impacts disproportionately around the world those living downstream who are most vulnerable to dam releases during flood events.

So, when considered in context, we believe our work is actually more suited for a water/hydrology journal rather than a remote sensing journal. Remote sensing is one of the many assets and tools employed here as a means to an end to seek answers to the following research questions:

*'How well can we apply a satellite remote sensing and model-based framework for near real-time monitoring of the dynamically changing state of hydropower reservoirs in mountainous and high precipitation regions?'*

*'With what certainty can such a modelling framework capture what transpired during the flooding event?'*

**Reviewer point 1** *: In my opinion, the current work fits better in journals about remote-sensing, and could be more appreciated there. Also, I think the current research should focus more on the long-term and sustainable water resources management (e.g., by adjusting the long-term operational rules of each reservoir to the climatic dynamics of the area and to the so-called Hurst phenomenon and global warming), and to the flood risk management (e.g., whether the operational rules over the last years seem to overestimate or underestimate or perfectly estimate the potential risk from flood events). I believe that it is difficult now for this tool to become operational for short-term predictions and flood management strategies (the reasons are explained in the next comments).*

**Our response**: We thank the reviewer's comments. Perhaps we may not have made the innovative aspects clearer in our earlier submission regarding regulated river basin management in mountainous basins with high precipitation. So, we now ask the reviewer to refer to our earlier response to the Reviewer#1's general assessment regarding the suitability of the work in the HESS journal. We provided a more detailed response about this to Reviewer#1 that this reviewer should be aware of. Also, we now articulate the findings and insights of our study that are generalizable and scalable for mountainous basins around the world in the conclusion section of the revised paper.

The key findings can be summarized as follows:

1) In mountainous, coastal and fast response basins, RAT3.0 was found to be able to track the temporal trends of the reservoir state with good accuracy. However, tracking reservoir storage changes at the highest frequency and accuracy is more important for such cases. Herein, we argue that the SWOT mission with the suite of nadir altimeters to track reservoir elevations will play a key role.

2) Given that RAT is model agnostic, mountainous regions require improved and better calibrated hydrologic models or reservoir inflow. In particular, the routing scheme requires attention as the area draining into the very upstream reservoirs is quite small in such highly mountainous basins where the dams are often at the edge of the boundary. This is where strong engagement from local partnering agencies to improve the calibration of the model (VIC

in our case of RAT 3.0) is critical. Fortunately for Kerala, we are already engaged with Kerala Centre for Water Resources Development and Management (CWRDM) and Kerala State Electricity Board (KSEB) who have agreed to help address this issue.

3) Because of perennial high cloud cover in such regions around the world with hydropower dams, microwave/radar-based satellite sensors are more critical and lay a central role in tracking reservoir state. We recommend that SWOT KaRIN sensor with as many radar altimeters (Sentinel 3A, 3B, 6, SWOT altimeter) be used for tracking reservoir storage change as accurately and frequently as possible for such regions.

While we acknowledge the importance of a study that focuses on the long-term aspects of flood management, global warming, and the long-term persistence of trends in hydrological time series data (Hurst phenomenon), we would like to stress that the current work was never intended to tackle such a problem in the first place. This study primarily focuses on answering the question of how effective a combined hydrological modelling and satellite remote sensing framework is in tracking reservoir states in high precipitation and steep topography regions and what insights are revealed in answering our overarching research question. Such an effort is missing in today's literature to the best of our knowledge. The study has shown that such a tool, herein the Reservoir Assessment Tool (RAT), is in-fact effective in tracking the dynamic state of the reservoir in such conditions and can contribute significantly to the short-term planning required for flood moderation. The downsides and limitations of the framework have also been explicitly highlighted to give an unbiased view of the effectiveness.

We also have operationalized RAT for Kerala for two agencies (Centre for Water Resources Development and Management -CWRDM, and Kerala State Electricity Board – KSEB) at http://depts.washington.edu/saswe/kerala wherein the system is now being co-developed further with agency provided data. So the reviewer is incorrect in their assessment that that "*it is difficult now for this tool to become operational for short-term predictions and flood management strategies.*"

We would also like to highlight that this study, along with other publications regarding the RAT framework sets a fundamental base from which further studies such as that recommended by the Reviewer can be carried out. A multi-decadal and global study exploring the long-term effects and trends in hydroelectric power demand and flood risk is actually one of our planned work using the RAT tool.

*Reviewer point 2 : The temporal resolution of the presented method, which is based on satellites, is above 10 days. I think this is somehow coarse to determine the contribution of the reservoirs to flood risk, and especially flash floods. Please consider discussing this issue and limitation, adjust the current research and results (e.g., how is the intermediate flood events are taken into account in the current research), and propose possible solutions (for example, monitoring instruments to measure hourly/daily the water-stage could be a necessary condition for a reservoir to be entered in the RAT3.0 flood management system).*

**Our response**: Thank you for the comment. The reviewer has mistakenly noted the temporal resolution of RAT3.0 as being above 10 days. In its current implementation, RAT3.0 has a temporal resolution of 1-5 days (average 2.33 days) for surface area, storage change and outflow estimations. This is achieved through a combination of multiple satellite sensors (Landsat 8, Landsat 9, Sentinel-1, 2) that are both optical and SAR based (methodology explained in Figure 6 in the manuscript). Inflow into the reservoir is modelled at the frequency of 1 day. We strongly believe that such a frequency is capable of providing extremely valuable insights into the reservoir states for most flood events that evolve very fast in mountainous basins such as Kerala. This 1-5 day frequency will only keep improving with the advent of newer satellite missions, which will be continuously added to the RAT framework for tracking fast response flood events in mountainous regions. We also feel that the idea of excluding reservoirs not measuring data at hourly/daily rates would limit the total number of reservoirs by a large margin and goes against the open and transparent nature of RAT.

Again, just to remind the reviewer – our RAT 3.0 operational set up for Kerala at http://depts.washington.edu/saswe/kerala is now an active application for two water agencies of the state of Kerala (CWRDM and KSEB).

*Reviewer point 3: Please mention possible additional sources of inflow and outflow that are not easily traced by the satellites, such as water-losses or uptakes or overflow through weirs, etc.*

**Our response**: This is a good recommendation, and we will add the possible source of additional inflow and outflow to the revised manuscript that RAT cannot track in its current state.

*Reviewer point 4: Please further explain how the surface water difference is translated to volume for the water-balance model since the surface elevation is not known or cannot be seen from the satellite for the submerged part.*

**Our response**: The translation of the surface water difference into volume for the water-balance model has been explained in Section 3.2.3 of the manuscript. RAT utilizes area-elevation relationships curves (AEC) derived from the digital elevation model data from the SRTM satellite mission as per the methodology outlined in the paper by Biswas et al. (2021). Figure below shows an example of AEC, which can also be derived from other sources, such as topographic surveys. If the AEC curve is generated using SRTM mission, it is extrapolated below the waterline that existed during Feb 2000 when SRTM mission flew, by mathematically fitting the SRTM observed DEM values. Using this AEC, the surface area of a reservoir at any given time can be mapped in accordance to its elevation and vice versa (from elevation to surface) as shown in the figure below. Two successive such surface area readings, then allow us to compute the storage change delta S, using the trapezoidal approximation shown below.

[Figure]

[Figure]

$$\Delta S = A_{avg.} * \Delta h = \frac{(A_2 + A_1)}{2} * (h_2 - h_1)$$

The exact algorithm that is employed in RAT to extract the area-elevation relationship is as per Fig1.

[Figure]

Fig1. Area-elevation relationship extraction from SRTM DEM Data (Biswas et al., 2021)

***Reviewer point 5:*** *It is mentioned in the conclusions that "This unexpectedly high amount of inflow, coupled with the insufficient flood cushioning provided caused the reservoirs to reach the maximum storage quickly and lose any ability to provide flood moderation.". However, this is not possible to know just by looking at the satellites; for example, maybe this action was necessary to face any unexpected high water-demand or drought during the summer. Maybe it is better to couple this methodology with the extracted water to satisfy the water demand for each area. In any case, I think this issue should be further discussed and analyzed.*

**Our response**: The conclusion cited was arrived at by looking at the reservoir water areas during the period of the flood event as shown in the example given in Fig2. It is to be noted here that the entire flood lies in the Monsoon season period in Kerala (July-Aug) and not in the Summer (February to June). Hence the reasoning behind the continuous filling of the reservoir being attributed to high water demand during the summer is ill-conceived. Furthermore, the conclusion that the continuous filling of the reservoir before the advent of the floods was a key reason in loss of flood moderation capability has been corroborated by existing literature on the 2018 Kerala Flood events as mentioned in the introduction section of the manuscript. Fig 3 shows the flood cushioning provided in June for the Kerala reservoir of Idamalayaar as identified from the RAT derived surface area. This is an intentional lowering of the reservoir level in early June in anticipation of the upcoming Monsoon rain in the month of July and August. We can look at the official operating rule curves for Idamalayaar, as given in Fig 4 to cross verify this. The actual rule curve for the reservoir shows an intentional lowering of the water levels in June which is what we are picking up in the Surface area time series.

[Figure]

Fig2. Reservoir surface area state during flood period and max surface area. The region in purple is the period of flood occurrence. The surface area just before the onset of the flood is already past 90% of the max surface area

[Figure]

Fig3. RAT derived Surface Area time series for Idamalayaar reservoir. Flood cushioning provided in June is highlighted in green.

[Figure]

Fig4. Official Rule curve for Idamalayaar – Kerala State Electricity Board (KSEB) report on Rule Curves of Major Reservoirs, 2019. Region in yellow showcases the intentional lowering of the water level in June

---

## Author Comment (AC3)

**Authors' Response to Reviewer#3**

We thank the reviewer for their timely review of our paper. We also thank the editor for securing the review in a timely manner. The overall assessment of the reviewer on the paper is that "*The main problem of this manuscript is lack of enough novelty as original research. However, it can be a case study paper (not original research).*

The complete review comment is as follows:

"*The main problem of this manuscript is lack of enough novelty as original research. However, it can be a case study paper (not original research). The approaches and methods are known and currently in use which means the authors need to add novelties to the method, if they would like to publish it as the original research paper. It is like an excellent technical report in the current form. My recommendation is to explore the methods better and adding some specific novelties to the methodology. In the current form, it can be seen (even from the title) that no significant insight is added*".

**Our response**:

We thank the reviewer's comments. We realize that we may not have made the innovative aspects of our study clearer in the original manuscript. Please note that we have provided a more detailed response in this regard to Reviewer#1 and we ask that this reviewer refers to our response to this same issue of 'novelty.'

The study primarily focuses on the application and effectiveness of a combined remote sensing and hydrological modelling framework in tracking reservoir operations during extreme weather events in high precipitation and steep topography regions. Kerala was chosen as the test bed for this purpose. The study has shown that such a tool, herein the Reservoir Assessment Tool (RAT), is in-fact effective in tracking the dynamic state of the reservoir in such conditions and can contribute significantly to the short-term planning required for flood moderation. The downsides and limitations of the framework have also been explicitly highlighted to give an unbiased view of the effectiveness.

Our work is also about the '***representation of water infrastructure in large scale hydrological models***' which is the HESS special issue theme. We pursue this by using state of the art data informatics solutions (e.g. cloud computing) and satellite remote sensing to solve the critical but less-understood problem of *flood preparedness in mountainous regions where hydropower dam operations exacerbate downstream flood risk.* Here the 'water infrastructure' is reservoir/dam that is explicitly accounted for in the modeling/prediction of flood events in fast response, high terrain basins where flood risk management by hydropower dams (that are generally designed to keep full supply level for power generation) are particularly challenging due to its traditional lack of transparency. To the best of our knowledge such work is missing in today's literature. So, we believe our work is innovative and a key contribution to the body of

knowledge as we have identified scalable or generalizable methods for dealing with the flood preparedness issue in similar environments around the world. This work can be scaled to other reservoir in such environments (Please refer Fig.1)

[Figure]

**Fig 1.** (also in the paper) – Regions in yellow show where the findings and lessons learned for RAT 3.0 application over Kerala during 2018 August floods (i.e., our HESS paper) can be applied around the world where flood risks appearing to be increasing due to the combination of climate change, energy production requirements and land use change.

Some of the key scalable findings are (which we plan to make clearer in our revised manuscript):

1) In mountainous, coastal, and fast response basins, RAT3.0 was found to be able to track the temporal trends of the reservoir state with good accuracy. However, tracking reservoir storage changes at the highest frequency and accuracy is more important for such cases. Herein, we argue that the SWOT mission with the suite of nadir altimeters to track reservoir elevations will play a key role.

2) Given that RAT is model agnostic, mountainous regions require improved and better calibrated hydrologic models or reservoir inflow. In particular, the routing scheme requires attention as the area draining into the very upstream reservoirs is quite small in such highly mountainous basins where the dams are often at the edge of the boundary. This is where strong engagement from local partnering agencies to improve the calibration of the model (VIC in our case of RAT 3.0) is critical. Fortunately for Kerala, we are already engaged with Kerala Centre for Water Resources Development and Management (CWRDM) and Kerala State Electricity Board (KSEB) who have agreed to help address this issue.

3) Because of perennial high cloud cover in such regions around the world with hydropower dams (see Fig 1 yellow regions), microwave/radar based satellite sensors is more critical and lay a central role in tracking reservoir state. We recommend that SWOT KaRIN sensor with as many

radar altimeters (Sentinel 3A, 3B, 6, SWOT altimeter) be used for tracking reservoir storage change as accurately and frequently as possible for such regions around the world identified in yellow in Fig 1.

In summary, we therefore argue that our work is more than just a technical report of RAT for Kerala. Although we had articulated the novelty of the work in lines 28-115, we plan to integrate a concise summary of the above in the introduction section of the paper to make it clearer how our work makes a contribution to the body of knowledge.